# Decoding the function of Atg13 phosphorylation reveals a role of Atg11 in bulk autophagy initiation

Anuradha Bhattacharya[1,2,3], Raffaela Torggler[1,2], Wolfgang Reiter [4,5], Natalie Romanov[4,8], Mariya Licheva [1,2], Akif Ciftci[1,2], Muriel Mari [6], Lena Kolb[4], Dominik Kaiser[1], Fulvio Reggiori [6], Gustav Ammerer[4], David M Hollenstein [1,4,5✉] & Claudine Kraft [1,7✉]

## Abstract

**Autophagy is initiated by the assembly of multiple autophagy-related proteins that form the phagophore assembly site where autophagosomes are formed. Atg13 is essential early in this process, and a hub of extensive phosphorylation. How these multiple phosphorylations contribute to autophagy initiation, however, is not well understood. Here we comprehensively analyze the role of phosphorylation events on Atg13 during nutrient-rich conditions and nitrogen starvation. We identify and functionally characterize 48 in vivo phosphorylation sites on Atg13. By generating reciprocal mutants, which mimic the dephosphorylated active and phosphorylated inactive state of Atg13, we observe that disrupting the dynamic regulation of Atg13 leads to insufficient or excessive autophagy, which are both detrimental to cell survival. We furthermore demonstrate an involvement of Atg11 in bulk autophagy even during nitrogen starvation, where it contributes together with Atg1 to the multivalency that drives phase separation of the phagophore assembly site. These findings reveal the importance of post-translational regulation on Atg13 early during autophagy initiation, which provides additional layers of regulation to control bulk autophagy activity and integrate cellular signals.**

**Keywords** Autophagy; PAS Formation; Atg1 Kinase Complex; Atg13; Atg11
**Subject Categories** Autophagy & Cell Death; Post-translational Modifications & Proteolysis; Signal Transduction

## Introduction

Eukaryotic cells respond to nutrient starvation by induction of bulk autophagy, a catabolic process that non-selectively degrades and recycles bulk cytosolic components. During bulk autophagy, a membrane structure is formed de-novo, expanded, and sealed to engulf a part of the cytosol in a double-membrane vesicle termed autophagosome. Fusion of the autophagosome with compartments of the endolysosomal system leads to the degradation of the inner autophagic membrane and the entrapped cytosolic material. The thereby generated molecular building blocks are exported into the cytosol and become available to the cell once again (Hollenstein and Kraft, 2020).

Under nutrient-rich conditions, bulk autophagy is inhibited by the action of the nutrient sensor Tor kinase complex 1 (TORC1) (Papinski and Kraft, 2016; Noda and Ohsumi, 1998) TORC1 acts on the autophagy machinery by hyper phosphorylating Atg13, which prevents the assembly of Atg13 with autophagy-related (Atg) factors and thereby pathway initiation. When starvation inhibits TORC1 activity, Atg13 undergoes rapid dephosphorylation, allowing its stable assembly with the autophagy machinery to form the phagophore assembly site (PAS) and initiate autophagosome formation (Kamada et al, 2000).

Atg13 plays a crucial role in autophagy by forming a complex with Atg1, the homolog of mammalian ULK1 and a key protein kinase involved in autophagy. This complex interacts with the dimeric Atg17-Atg29-Atg31 (Atg17[-Atg29-Atg31]) complex, resulting in the formation of an Atg1-Atg13-Atg17[-Atg29-Atg31] supramolecular assembly, known as the Atg1 complex, or ULK1 complex in mammals (Kabeya et al, 2009; Young et al, 2006). The multivalent interaction of Atg17 with Atg13 drives the phase separation of the Atg1 complex, thereby forming the early bulk autophagy PAS (Fujioka et al, 2020). The resulting liquidity of the PAS has been proposed to be critical for the subsequent recruitment of other Atg proteins, which promotes PAS assembly and initiates autophagosome formation.

Atg13 cross-links Atg17[-Atg29-Atg31] complexes via two distinct binding regions, termed LR (linking region, Atg13[359-389]) and BR (binding region, Atg13[424-436], Fig. 1A). In both of these regions, TORC1-dependent phosphorylation regulates the interaction with Atg17, on Ser379 and Ser428/Ser429, respectively (Fujioka et al, 2014; Yamamoto et al, 2016). Similarly, five TORC1-dependent serine phosphorylations have been mapped in the Atg1-interaction region of Atg13 (MIM, Fig. 1A). A mutant that mimics the

[1]Institute of Biochemistry and Molecular Biology, ZBMZ, Faculty of Medicine, University of Freiburg, 79104 Freiburg, Germany. [2]Faculty of Biology, University of Freiburg, 79104 Freiburg, Germany. [3]Spemann Graduate School of Biology and Medicine (SGBM), University of Freiburg, 79104 Freiburg, Germany. [4]Department for Biochemistry and Cell Biology, University of Vienna, Center for Molecular Biology, Vienna Biocenter Campus (VBC), Dr. Bohr-Gasse 9, 1030 Vienna, Austria. [5]Mass Spectrometry Facility, Max Perutz Labs, Vienna Biocenter Campus (VBC), Dr. Bohr-Gasse 7, Vienna, Austria. [6]Department of Biomedicine, Aarhus University, Ole Worms Allé 4, 8000 Aarhus C, Denmark. [7]CIBSS - Centre for Integrative Biological Signalling Studies, University of Freiburg, 79104 Freiburg, Germany. [8]Present address: Department of Molecular Sociology, Max Planck Institute of Biophysics, Frankfurt, Germany. ✉E-mail: david.hollenstein@univie.ac.at; kraft@biochemie.uni-freiburg.de

phosphorylated, inactive state of Atg13 (Atg13[5SD]), however, only caused a 25% bulk autophagy defect; and the non-phosphorylatable, reciprocal Atg13[5SA] mutant was unable to increase the affinity of Atg13 to Atg1 (Fujioka et al, 2014). Another study mutated eight phosphorylation sites distributed all over Atg13. This Atg13[8SA] mutant resulted in functional bulk autophagy and was even capable of inducing bulk autophagy under nutrient-rich conditions when overexpressed, suggesting that it can bypass the inhibitory activity of TORC1. However, the reciprocal allele Atg13[8SD] which mimics the phosphorylated, inactive state, was unable to efficiently inhibit bulk autophagy (Kamada et al, 2010). As neither of these mutants were able to mimic reciprocal states, it remains unclear whether the dephosphorylation and phosphorylation of a specific set of residues on Atg13 is sufficient for the induction and inhibition of the autophagy pathway, and how other autophagy factors are influenced by these phosphorylation events. Moreover, a multitude of additional phosphorylation sites on Atg13 have been identified in vivo, but not functionally analyzed, further supporting the notion that Atg13 regulation by phosphorylation is far from being understood (Hu et al, 2019). We, therefore, conducted an extensive investigation into the effects of Atg13 phosphorylation. A total of 48 phosphorylation sites on Atg13 were identified in vivo by mass spectrometry analysis. Mutating most of these sites to non-phosphorylatable or phospho-mimetic residues resulted in a hyperactive or fully inhibited state of autophagy, respectively. Thus, Atg13 acts as a signaling hub that can promote or inhibit autophagy, depending on its phosphorylation state. Furthermore, a comprehensive mutational analysis uncovered the crucial contributions of Atg1 and Atg11 to the formation of the bulk autophagy PAS, contributing to the phase transition of the Atg1 complex. Collectively, these results provide compelling evidence for the critical involvement of numerous phosphorylation events on Atg13 in precisely regulating the initiation of bulk autophagy.

# Results

Atg13 is a heavily phosphorylated protein, whose phosphorylation status regulates bulk autophagy activity (Kamada et al, 2000). Despite numerous phospho-mapping studies, the functional significance of most phosphorylation events remains unknown. To address how Atg13 phosphorylation affects autophagy activity, we reinvestigated the effect of phospho-mimetic mutants in the Atg17-LR and Atg17-BR regions, on Ser379, Ser428, and Ser429 (Fig. 1A,B). A phospho-mimetic mutant of Ser428 and Ser429 (Atg13[S428D/S429D]) showed a severe defect in the autophagy flux, assessed by monitoring the vacuolar delivery of Pho8Δ60, in comparison to wild-type Atg13 (wt, Atg13[wt]) (Noda and Klionsky, 2009), in agreement with a previous report (Fujioka et al, 2014, Fig. 1C). To our surprise, a phosphomimetic mutant of Ser379, Atg13[S379D] showed almost no impact on autophagy activity in a wild-type background (Fig. 1D). This is in contrast to a previous report (Yamamoto et al, 2016), which, however, only analyzed this mutant in the absence of Atg11, a protein considered to mainly affect selective, but not bulk autophagy. Indeed, in atg11Δ cells, the Ser379D mutation showed an enhanced defect on the autophagic flux of about 50%, similar to previous reports (Fig. 1D, Yamamoto et al, 2016). Thus, Ser428 and Ser429 phosphorylation, but not Ser379 phosphorylation, appears sufficient to inhibit bulk

autophagic flux. These findings suggest that deletion of ATG11 in combination with other mutations of the core autophagy machinery can have synthetic defects on bulk autophagy. We, therefore, performed our further analysis of Atg13 in bulk autophagy in Atg11-expressing cells.

## Atg13 is a phosphorylation hub with a major impact on the autophagy flux

As many more phosphorylation sites on Atg13 than Ser379, Ser428, and Ser429 have been reported (Hu et al, 2019; Fujioka et al, 2014; Yamamoto et al, 2016), we reassessed Atg13 phosphorylation under nutrient-rich and starvation conditions (Dataset EV1). We mapped 48 phosphorylation sites by mass spectrometry, of which 36 were regulated by the nutrient status. Among these sites were also the previously reported sites Ser379, Ser428, and Ser429. Most regulated phosphorylation sites were present under nutrient-rich conditions and reduced or absent upon starvation, except for two sites that were reversely regulated (Fig. 1A,B).

To investigate the impact of these additional phosphorylation sites on the autophagic flux, we excluded Ser428 and Ser429 from our mutational analysis. These two serine residues are crucial for hydrogen bonding with Atg17 and therefore significantly affect the autophagy flux when mutated either to alanine or aspartate (Fig. 1C, Fujioka et al, 2014). By excluding them, we aimed to investigate the effect of the other phosphorylation sites independent of their known role in Atg17 binding. We created a phosphomimetic Atg13 mutant, Atg13[44D], in which the 44 sites phosphorylated under nutrient-rich conditions, excluding Ser428/Ser429, were mutated to phosphomimetic aspartate residues, but the two sites that were upregulated only upon starvation, S344 and S496, were mutated to non-phosphorylatable alanine. We also created a mutant that included the Ser428 and Ser429 to aspartate mutations and termed it Atg13[46D] (which also includes the S344A and S496A mutations among the 46 aspartate substitutions). Throughout this study, mutated versions of Atg13 were either stably integrated at the native ATG13 locus in the genome, or expressed from a centromeric plasmid containing the native ATG13 promoter in an atg13Δ deletion strain. Despite the many mutations, these mutants were stably expressed (Fig. 2A). The Atg13[46D] mutant showed a complete defect in bulk autophagy activity, as expected, but also the Atg13[44D] mutant was fully defective, suggesting that further phosphorylation sites in addition to Ser428 and Ser429 play important roles in autophagy regulation (Fig. 2B).

## The mutation of 46 in vivo phosphorylation sites on Atg13 enables mimicking reciprocal states of autophagy activity

If phosphorylation of Atg13 inhibits bulk autophagy, then dephosphorylation should promote the pathway. To test this possibility, we generated the reciprocal non-phosphorylatable mutant Atg13[44A] (44 alanine and 2 aspartate mutations). This mutant showed enhanced bulk autophagy flux compared to Atg13[wt]-expressing cells, suggesting that cells carrying Atg13[44A] undergo a hyperactive autophagic state (Fig. 2B). We also generated the reciprocal mutant Atg13[46A], which includes the S428 and S429 alanine mutations (46 alanine and 2 aspartate mutations). Co-immunoprecipitation of Atg17 with Atg1-protA demonstrated that the interaction between

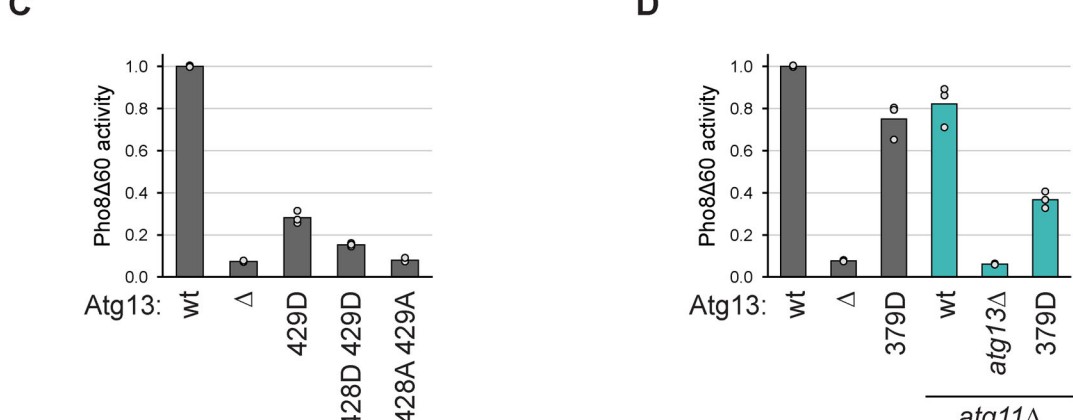

**Figure 1. Identification of multiple phosphorylations on Atg13 by mass spectrometry.**

(A, B) Cartoon and sequence of Atg13 depicting phosphorylation sites identified by mass spectrometry. Sites present under nutrient-rich conditions and downregulated upon starvation are depicted in red, and sites upregulated upon starvation are in green. Sites present irrespective of the nutrient state are shown in blue. LR, BR: Atg17 binding regions; MIM: Atg1 interaction region; V8: Vac8 binding region. (C) Atg13$^{wt}$ pho8Δ60, atg13Δ pho8Δ60, Atg13$^{429D}$ pho8Δ60, Atg13$^{428D429D}$ pho8Δ60 or Atg13$^{428AA429A}$ pho8Δ60 cells were starved for 4 h. Pho8Δ60 alkaline phosphatase activity was measured in three independent experiments. The values of each replicate (circles) and mean (bars) were plotted. All values were normalized to the mean Pho8Δ60 alkaline phosphatase activity of Atg13$^{wt}$ pho8Δ60 cells. (D) atg13Δ pho8Δ60, atg11Δ atg13Δ pho8Δ60, Atg13$^{wt}$ pho8Δ60, Atg13$^{379D}$ pho8Δ60, Atg13$^{wt}$ atg11Δ pho8Δ60 or Atg13$^{379D}$ atg11Δ pho8Δ60 cells were starved for 4 h. Pho8Δ60 alkaline phosphatase activity was measured in three independent experiments. The values of each replicate (circles) and mean (bars) were plotted. All values were normalized to the mean Pho8Δ60 alkaline phosphatase activity of Atg13$^{wt}$ pho8Δ60 cells. Source data are available online for this figure.

**A**

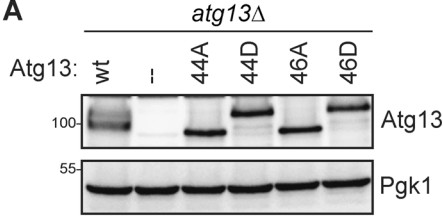

**D**

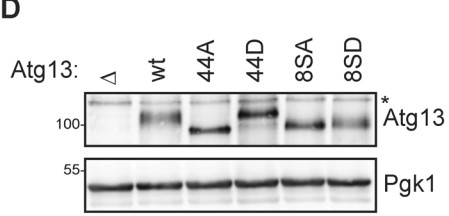

**B**

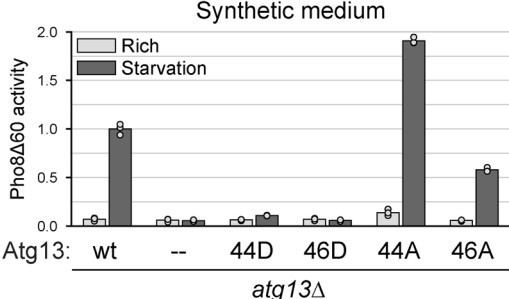

**E**

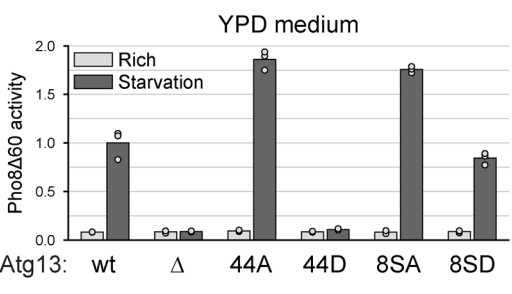

**C**

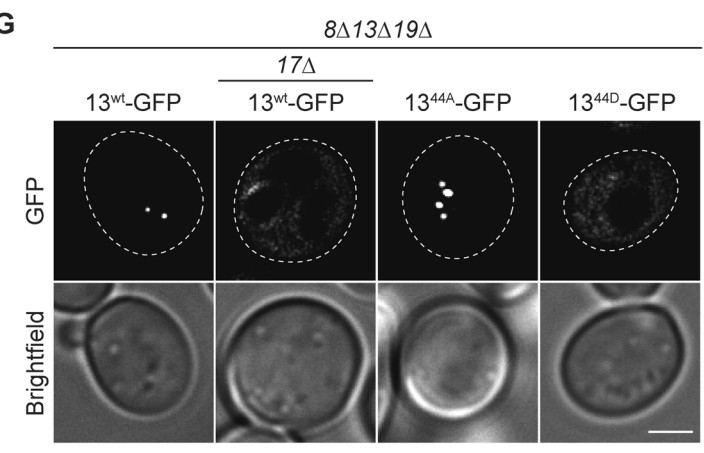

**F**

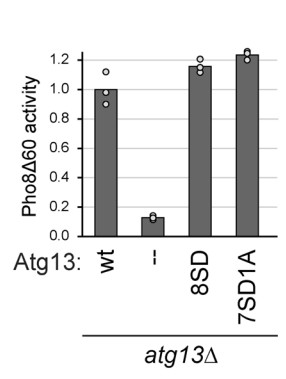

**G**

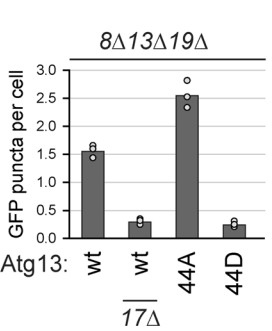

**Figure 2. The mutation of 46 in vivo phosphorylation sites on Atg13 enables mimicking reciprocal autophagy states.**

(A) $atg13\Delta$ $pho8\Delta60$ cells containing plasmid expressed Atg13$^{wt}$, Atg13$^{44A}$, Atg13$^{46A}$, Atg13$^{44D}$, Atg13$^{46D}$ or an empty plasmid were grown to log phase, and cell extracts were prepared by TCA precipitation. Expression levels of proteins were analyzed by anti-Atg13 and anti-Pgk1 western blotting. One representative experiment out of two is shown. (B) $atg13\Delta$ $pho8\Delta60$ cells containing plasmid expressed Atg13$^{wt}$, Atg13$^{44D}$, Atg13$^{46D}$, Atg13$^{44A}$, Atg13$^{46A}$ or empty plasmid were grown in synthetic medium to log phase and starved for 4 h. Pho8$\Delta$60 alkaline phosphatase activity was measured in three independent experiments. The values of each replicate (circles) and mean (bars) were plotted. All values were normalized to the mean Pho8$\Delta$60 alkaline phosphatase activity of cells expressing Atg13$^{wt}$. (C) Atg1-protA Atg17-myc $atg13\Delta$ cells containing plasmid expressed Atg13$^{wt}$, Atg13$^{46A}$, Atg13$^{46D}$, Atg13$^{44A}$, Atg13$^{44D}$ or an empty plasmid were starved for 1 h. Atg1-protA was immunoprecipitated using IgG beads, and immunoprecipitates were analyzed by anti-protein A (PAP), anti-Atg13, and anti-myc western blotting. Asterisk: non-specific band. One out of four independent biological replicates is shown. (D) $atg13\Delta$ $pho8\Delta60$, Atg13$^{wt}$ $pho8\Delta60$, Atg13$^{44A}$ $pho8\Delta60$, Atg13$^{44D}$ $pho8\Delta60$, Atg13$^{8SA}$ $pho8\Delta60$ or Atg13$^{8SD}$ $pho8\Delta60$ cells were grown to log phase, and cell extracts were prepared by TCA precipitation. Expression levels of proteins were analyzed by anti-Atg13 and anti-Pgk1 western blotting. Asterisk: non-specific band. One out of two independent biological replicates is shown. (E) Atg13$^{wt}$ $pho8\Delta60$, $atg13\Delta$ $pho8\Delta60$, Atg13$^{44A}$ $pho8\Delta60$, Atg13$^{44D}$ $pho8\Delta60$, Atg13$^{8SA}$ $pho8\Delta60$ or Atg13$^{8SD}$ $pho8\Delta60$ cells were grown in YPD medium to log phase and starved for 4 h. Pho8$\Delta$60 alkaline phosphatase activity was measured in three independent experiments. The values of each replicate (circles) and mean (bars) were plotted. All values were normalized to the mean Pho8$\Delta$60 alkaline phosphatase activity of Atg13$^{wt}$ $pho8\Delta60$ cells. (F) $atg13\Delta$ $pho8\Delta60$ cells containing plasmid expressed Atg13$^{wt}$, Atg13$^{8SD}$, Atg13$^{7SD1A}$, or an empty plasmid were starved for 4 h. Pho8$\Delta$60 alkaline phosphatase activity was measured in three independent experiments. The values of each replicate (circles) and mean (bars) were plotted. All values were normalized to the mean Pho8$\Delta$60 alkaline phosphatase activity of cells expressing Atg13$^{wt}$. (G) $atg8\Delta$ $atg13\Delta$ $atg19\Delta$ or $atg8\Delta$ $atg13\Delta$ $atg17\Delta$ $atg19\Delta$ cells containing plasmid expressed Atg13$^{wt}$-GFP, Atg13$^{44A}$-GFP or Atg13$^{44D}$-GFP were starved for 30 min and analyzed by fluorescence microscopy. GFP puncta per cell were quantified in three independent experiments. For each strain and replicate at least 100 cells were analyzed. Scale bar: 2 μm. Source data are available online for this figure.

them relied entirely on the presence of Atg13 (Fig. 2C). While Atg13$^{44A}$ coprecipitated with Atg1-protA similar to Atg13 wild-type, co-precipitation of Atg13$^{46A}$ was reduced. The Atg13$^{46A}$ mutant contains the S428A and S429A mutations, two sites that have previously been reported to be required for binding to Atg17 (Fujioka et al, 2014), therefore it appears likely that the reduced co-precipitation of Atg13$^{46A}$ with Atg1 is due to less stable PAS formation of Atg13$^{46A}$. These findings show that the 46 in vivo mapped phosphorylation sites mutated in Atg13$^{44A}$ and Atg13$^{44D}$ (44A: 44 sites mutated to A and 2 sites to D; 44D: 44 sites mutated to D and 2 sites to A) can mimic the autophagy-active and autophagy-inhibited states, respectively.

It has been reported previously that a mutant allele of Atg13 (Atg13$^{8SA}$), with eight non-phosphorylatable residues, exhibited hyperactivity in bulk autophagy. Four of the eight mutated residues had been mapped in vivo (Ser437, Ser438, Ser646, and Ser649), whereas the other four had been assigned by sequence similarities without in vivo verification (Ser348, Ser496, Ser535, and Ser541, (Kamada et al, 2010). All eight sites were also identified in our in vivo phospho-mapping analysis, but one site, Ser496, was phosphorylated instead of dephosphorylated upon starvation and therefore mutated in a reverse manner in the Atg13$^{44A}$ and Atg13$^{44D}$ mutants (Fig. 1B). Despite this difference, we compared the previously reported non-phosphorylatable Atg13$^{8SA}$ and the reciprocal phospho-mimetic Atg13$^{8SD}$ mutants to our Atg13$^{44A}$ and Atg13$^{44D}$ mutants, all expressed at endogenous levels. While the Atg13$^{8SA}$ mutant showed a similar increase in the autophagy flux compared to Atg13$^{44A}$, the Atg13$^{8SD}$ mutant remained proficient in autophagy and was unable to mimic the reciprocal, inhibited state (Fig. 2D,E) We also created a mutant of these 8 sites, in which Ser496 was mutated reciprocally (7SD1A), as in the Atg13$^{44D}$ mutant. However, also this mutant was unable to mimic the inhibited state (Fig. 2F), suggesting that these eight sites are not sufficient to regulate Atg13 activity.

Together, these findings suggest that multiple phosphorylation events on Atg13, beyond phosphorylation of Ser428 and Ser429, regulate bulk autophagy activity. Remarkably, the full non-phosphorylatable allele was capable of inducing a hyperactive autophagy state, while the reciprocal, phospho-mimetic allele completely inhibited bulk autophagy.

## Early PAS formation requires Atg13 dephosphorylation

Next, we asked if PAS formation is affected by our Atg13 phosphorylation mutants. We expressed GFP-tagged Atg13$^{wt}$, Atg13$^{44A}$ or Atg13$^{44D}$ in $atg8\Delta$ $atg13\Delta$ $atg19\Delta$ cells. To monitor only bulk autophagy, we used cells not expressing the cytoplasm-to-vacuole transport (Cvt) pathway cargo-receptor Atg19, which lack Cvt pathway-induced PAS formation (Shintani et al, 2002; Scott et al, 2001). Furthermore, we used an $atg8\Delta$ background, which results in cells being unable to form autophagic membranes, thereby stalling PAS progression and preventing its turnover. This experimental setup allows the specific analysis of early PAS formation during bulk autophagy in a comparable manner between different mutants (Hollenstein et al, 2019). We observed that cells expressing Atg13$^{44A}$ formed more PAS structures than Atg13$^{wt}$ cells, but the Atg13$^{44D}$ mutant was completely defective in PAS formation, suggesting that extensive dephosphorylation of Atg13 is required to induce the assembly of the early PAS upon starvation (Fig. 2G).

## Atg13$^{44A}$ induces bulk autophagy under nutrient-rich conditions without affecting TORC1 signaling

When analyzing the autophagic activity, we detected some autophagy flux also under nutrient-rich conditions in the Atg13$^{44A}$ mutant, but not the Atg13$^{wt}$ cells (Fig. 2B). This slight induction was significant ($p = 0.03$), but only visible when cells were grown in synthetic SD medium but not in YPD medium (Fig. 2E). As cells more rapidly divide in YPD than in SD medium, and hardly any division occurs upon starvation (Fig. 3A), we speculated that the vacuolar Pho8$\Delta$60 signal might be out-diluted by the more frequent cell divisions when cells are grown in YPD, thereby masking the basal bulk autophagy activity of the Atg13$^{44A}$ mutants. If Atg13$^{44A}$ cells induce bulk autophagy under nutrient-rich conditions, then inhibiting cell division should increase the measurable Pho8$\Delta$60 levels. To test this possibility, we inhibited cell division by treatment with hydroxyurea (HU). HU is an inhibitor of the ribonucleotide reductase, which causes decreased levels of dNTPs and therefore decreased DNA replication, resulting in a cell cycle arrest in S-phase (Fig. 3A, Rosebrock, 2017). Importantly, the cell

## A

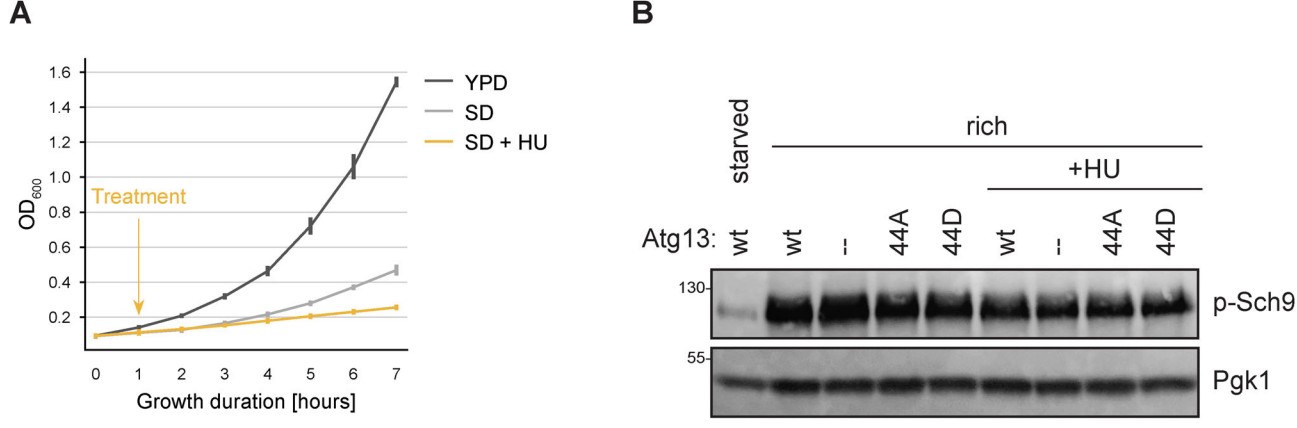

## B

## C

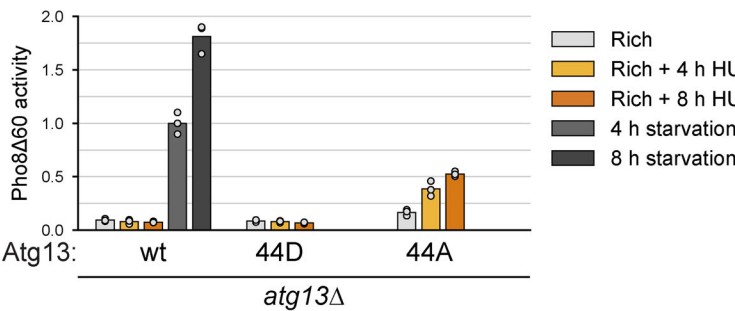

## D

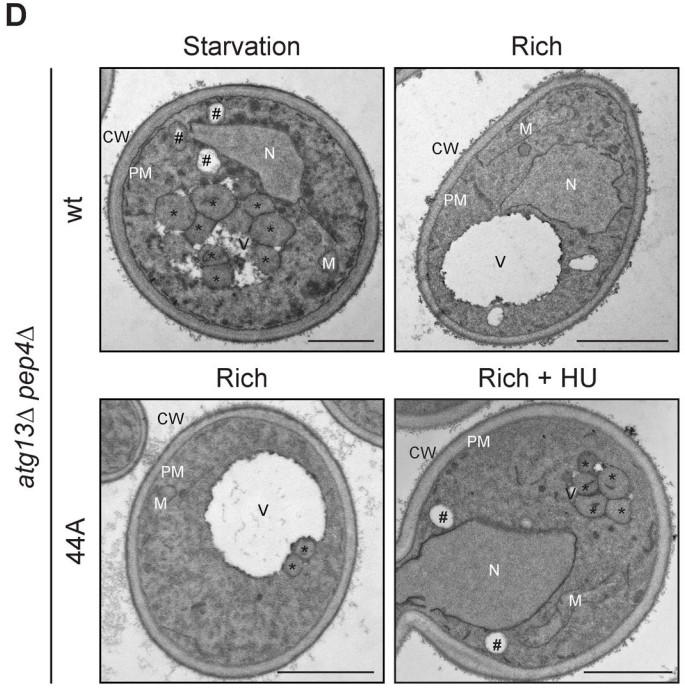

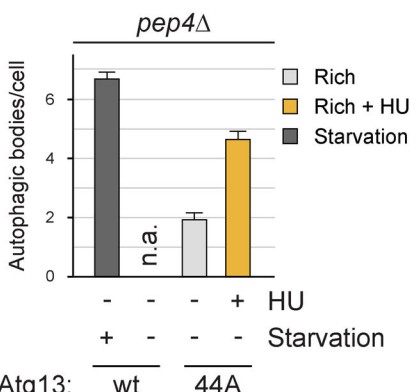

Figure 3. Atg13[44A] induces bulk autophagy under nutrient-rich conditions without affecting TORC1 signaling.

(A) pho8Δ60 cells containing no plasmid or an empty URA expression plasmid were grown in log phase in YPD or SD-URA medium, cells in SD-URA were treated with hydroxyurea (HU) or mock treated. Cell growth was monitored by OD_{600} measurement in three independent experiments. The line plot represents the replicate means, with error bars indicating twice the standard deviation. (B) Atg13[wt], Atg13[44A], Atg13[44D], or atg13Δ cells were grown in log phase and treated with hydroxyurea (HU) for 4 h or starved for 4 h, as indicated, and cell extracts were prepared by TCA precipitation. TORC1 activity was analyzed by anti-phosphoT737-Sch9 western blotting. One representative experiment out of two is shown. (C) atg13Δ pho8Δ60 cells containing plasmid expressed Atg13[wt], Atg13[44A,] or Atg13[44D] were grown in log phase, followed by starvation or treatment with hydroxyurea (HU) for 4 h and 8 h, as indicated. Pho8Δ60 alkaline phosphatase activity was measured in three independent experiments. The values of each replicate (circles) and mean (bars) were plotted. All values were normalized to the mean Pho8Δ60 alkaline phosphatase activity of cells expressing Atg13[wt]. (D) atg13Δ pep4Δ cells containing plasmid expressed Atg13[wt] or Atg13[44A] were grown in log phase, and starved for 1.5 h or treated with hydroxyurea (HU) for 4 h. Cells were analyzed by transmission electron microscopy. Representative electron micrographs are shown. CW, cell wall; M, mitochondria; N, nucleus; PM, plasma membrane; V, vacuole; *, autophagic bodies; #, lipid droplets. The number of autophagic bodies per vacuole section was quantified in three independent technical replicates. For each condition 180 vacuole profiles were analyzed. Scale bar: 1 µm. The quantification is represented as a bar plot, with bars representing replicate means and error bars indicating the standard error. n.a., not applicable. Source data are available online for this figure.

cycle arrest caused by HU treatment does not result in TORC1 inhibition, which we addressed by monitoring the phosphorylation status of the TORC1 substrate Sch9 (Fig. 3B, Urban et al, 2007). Upon four hours of HU treatment, we observed a strong increase in Pho8Δ60 activity under nutrient-rich conditions in Atg13[44A]-containing cells, which further increased upon prolonged HU treatment (Fig. 3C). Importantly, HU treatment did not result in elevated Pho8Δ60 activity under nutrient-rich conditions in Atg13[wt] cells, showing that the HU-induced cell cycle arrest by itself does not trigger bulk autophagy. In summary, these findings support that the non-phosphorylatable Atg13[44A] mutant is sufficient to trigger bulk autophagy also under nutrient-rich conditions when TORC1 is active.

To compare if bulk autophagosomes formed in Atg13[44A] mutants under nutrient-rich conditions are morphologically similar to those formed in Atg13[wt] cells upon starvation, we analyzed these cells by transmission electron microscopy. We visualized the autophagic bodies in the vacuolar lumen of vacuolar protease-deficient pep4Δ cells (Guimaraes et al, 2015). Autophagic bodies accumulated in the vacuole of Atg13[wt] pep4Δ cells upon starvation, but not under nutrient-rich conditions, as expected (Fig. 3D). In contrast, Atg13[44A] pep4Δ cells accumulated some autophagic bodies under nutrient-rich conditions. This number was further increased upon HU treatment, consistent with the increased Pho8Δ60 activity we observed under these conditions. We observed no obvious differences in morphology between the autophagic bodies of Atg13[wt] and Atg13[44A] mutant cells, suggesting that Atg13[44A] mutants are capable to form bulk autophagosomes under nutrient-rich conditions, similar to the ones generated upon starvation in wild-type cells.

## Bulk autophagosome formation under nutrient-rich conditions is facilitated by Atg11

Several autophagy factors, such as Atg1 and Atg13, are required for both selective and bulk autophagy pathways, whereas other factors, such as cargo receptors (e.g. Atg19, the selective autophagy receptor for the Cvt pathway), are required only for their corresponding selective type of autophagy pathway (Kamada et al, 2000). The scaffold proteins Atg11 and Atg17 were initially thought to function exclusively in different autophagy pathways, with Atg11 being required for the selective pathways and Atg17 for the non-selective bulk process upon nutrient starvation (Kamada et al, 2000). We, therefore, tested if bulk autophagy under nutrient-rich conditions in Atg13[44A] mutants depended on the same machinery

as bulk autophagy induced under starvation in Atg13[wt] cells. Similar to bulk autophagy during starvation, Atg17 was found to be necessary under rich conditions in Atg13[44A] mutants, while Atg19 was dispensable (Fig. 4A,B, Kabeya et al, 2005; Cheong et al, 2005; Chang and Huang, 2007). Unexpectedly, and in contrast to the situation under nitrogen starvation, Atg13[44A] mutants required Atg11 for efficient bulk autophagy induction when nutrients were available. This is in line with recent reports that Atg11 can contribute to bulk autophagosome formation under certain conditions, such as glucose or phosphate starvation (Adachi et al, 2017; Yao et al, 2020; Yokota et al, 2017). Co-immunoprecipitation experiments furthermore revealed increased levels of Atg13[44A] and Atg11 bound to Atg1 under nutrient-rich conditions (Fig. 4C).

Together, these findings suggest that mimicking the dephosphorylated state of Atg13 is sufficient to bypass TORC1 signaling and induce bulk autophagy under nutrient-rich conditions, which requires Atg17 and is facilitated by Atg11.

## Constitutive high bulk autophagy activity is detrimental to survival during nitrogen starvation

As the Atg13[44A] mutant showed increased autophagy activity also under starvation, we asked if this enhanced autophagy flux affected cell survival. Indeed, dilution spotting on nutrient-rich YPD plates after 14 days of starvation resulted in a major defect of the Atg13[44A] mutant but not Atg13[wt] cells to survive and resume growth (Fig. 4D). When monitoring the autophagy flux during the first 24 h, autophagy stagnated around eight to twelve hours in Atg13[wt] cells but continued to increase in Atg13[44A] mutants (Fig. 4E). Such a decrease in autophagy activity upon prolonged starvation in wild-type cells is in agreement with previous studies and has been proposed to be mediated by Atg1-dependent re-phosphorylation of Atg13 (Kira et al, 2021). As the Atg13[44A] mutant showed increasing autophagy flux after 12 h, it appears that this mutant cannot be turned off efficiently and remains constitutively active, ultimately resulting in cell death. These findings support the notion that during starvation too little autophagy but also too much autophagy is detrimental for cell survival, underlining the need for fine-tuned and balanced autophagy regulation.

## Atg13's C-terminal phosphorylations impact Atg1 interaction in trans and control PAS formation

Previous studies have either addressed Atg13 phosphorylation in the LR and BR region that regulate Atg17 binding (Ser379 and

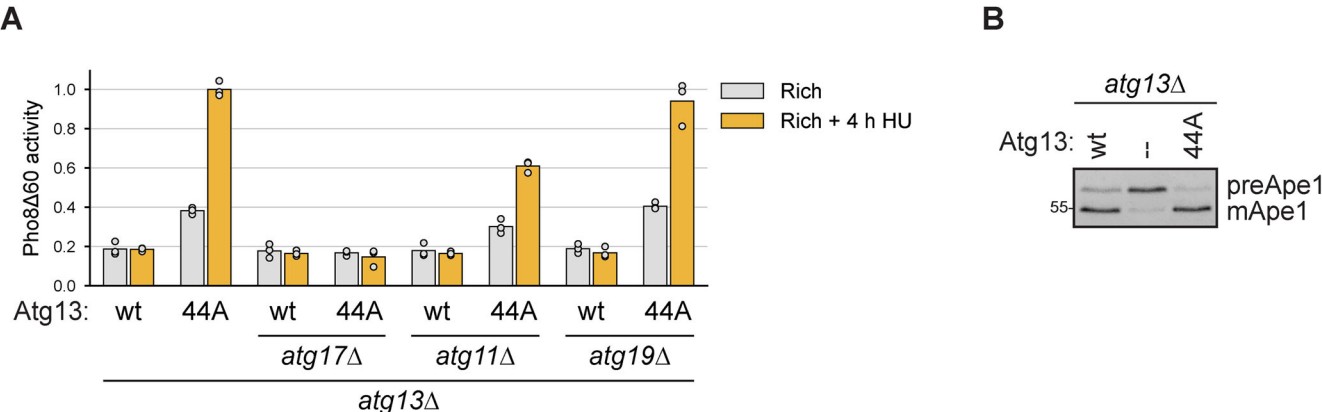

**A**

**B**

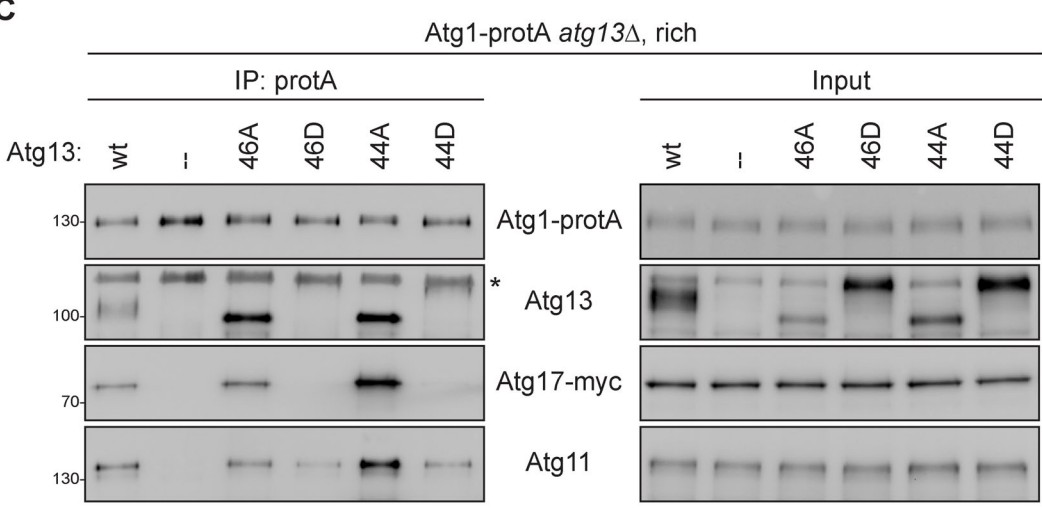

**C**

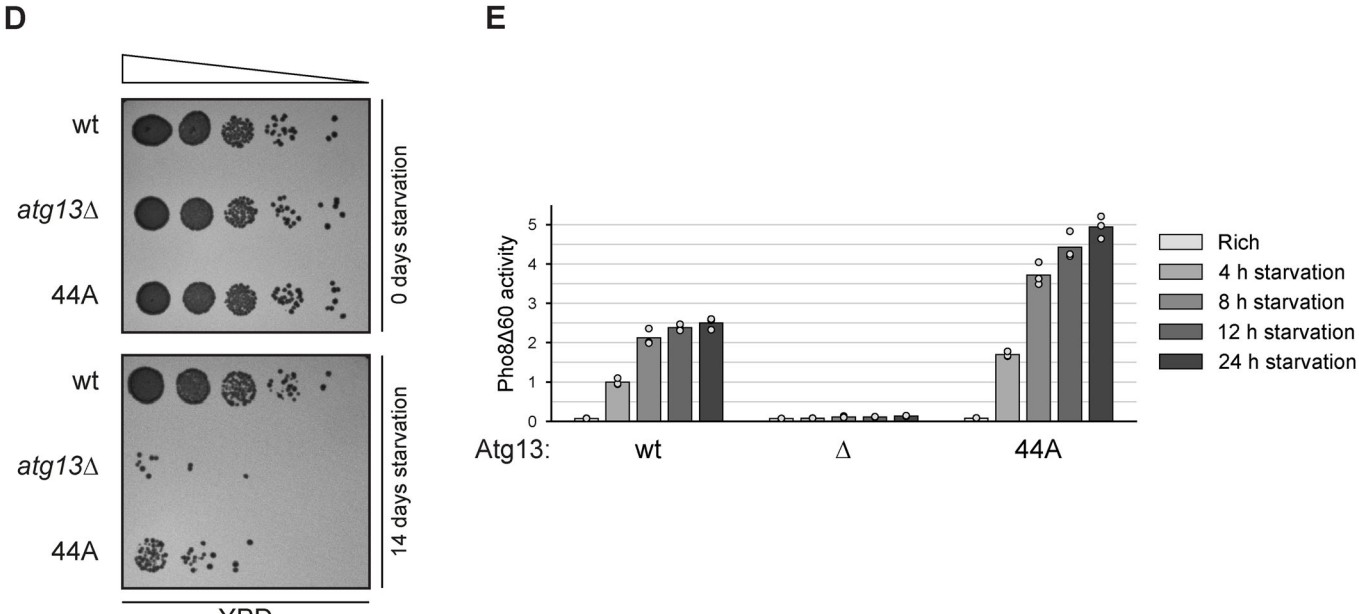

**D**

**E**

**Figure 4. Bulk autophagosome formation under nutrient-rich conditions is facilitated by Atg11.**

(A) *atg13Δ pho8Δ60, atg13Δ atg17Δ pho8Δ60, atg11Δ atg13Δ pho8Δ60,* and *atg13Δ atg19Δ pho8Δ60* cells containing plasmid expressed Atg13[wt] or Atg13[44A] were grown in log phase and treated with hydroxyurea (HU) for 4 h. Pho8Δ60 alkaline phosphatase activity was measured in three independent experiments. The values of each replicate (circles) and mean (bars) were plotted. All values were normalized to the mean Pho8Δ60 alkaline phosphatase activity of cells expressing Atg13[44A]. (B) *atg13Δ pho8Δ60* cells containing plasmid expressed Atg13[wt], Atg13[44A], or an empty plasmid were grown to log phase, and cell extracts were prepared by TCA precipitation. Samples were analyzed by anti-Ape1 western blotting. One out of two independent biological replicates is shown. (C) Atg1-protA Atg17-myc *atg13Δ* cells containing plasmid expressed Atg13[wt], Atg13[46A], Atg13[46D], Atg13[44A], Atg13[44D] or an empty plasmid were grown in log phase. Atg1-protA was immunoprecipitated using IgG beads, and immunoprecipitates and input extracts were analyzed by anti-protein A (PAP), anti-Atg13, anti-myc, and anti-Atg11 western blotting. Asterisk: non-specific band. One out of four independent biological replicates is shown. (D) Atg13[wt], *atg13Δ* and Atg13[44A] cells were starved for 0 or 14 days and spotted in serial dilutions onto YPD plates. One representative experiment out of three is shown. (E) Atg13[wt] *pho8Δ60*, Atg13[44A] *pho8Δ60,* and *atg13Δ pho8Δ60* cells were grown in log phase and starved for 4 h, 8 h, 12 h, and 24 h as indicated. Pho8Δ60 alkaline phosphatase activity was measured in three independent experiments. The values of each replicate (circles) and mean (bars) were plotted. All values were normalized to the mean Pho8Δ60 alkaline phosphatase activity of Atg13[wt] *pho8Δ60* cells after 4 h of starvation. Source data are available online for this figure.

Ser428/Ser429, Yamamoto et al, 2016; Fujioka et al, 2014) or phosphorylations in the MIM domain, which regulate Atg1 interaction (Ser484, Ser494, Ser496, Ser515, Ser517, Fujioka et al, 2014). All of these phosphorylations are located in the middle region of Atg13 (Fig. 1A). The Atg13[44D] mutant also displayed a severely reduced Atg1 and Atg17 binding (Fig. 2C), however, the mutated residues of Atg13[44D] are not only located in the middle region but are distributed over a large part of Atg13 (Fig. 1A). To address if phosphorylations outside the middle region affect the bulk autophagy flux, we split the mutated residues of Atg13[44A] and Atg13[44D] into N-terminal, middle, and C-terminal sites, and generated non-phosphorylatable and phosphomimetic mutants of the individual regions (Fig. 5A). In all these mutants, Ser428 and Ser429 were not mutated, as any mutation of those sites strongly affects Atg17 interaction (Fig. 1C, Fujioka et al, 2014). Non-phosphorylatable mutations in the middle region (Atg13[MA]) supported bulk autophagy to slightly higher levels than wild-type Atg13 (*p* value 0.00443, Fig. 5B), whereas the phospho-mimetic Atg13[MD] mutant was largely, but not completely defective (*p* value 0.00002, Fig. 5B). This is in contrast to Atg13[44A] and Atg13[44D], which showed a hyperactive and completely defective autophagy state (Fig. 2B), suggesting that Atg13 phosphorylation outside of the middle region contributes to the regulation of bulk autophagy. Next, we tested non-phosphorylatable and phospho-mimetic mutants of the N-terminal (Atg13[NA], Atg13[ND]) and the C-terminal regions (Atg13[CA], Atg13[CD]). Whereas N-terminal mutations did neither promote nor inhibit autophagy, the Atg13[CA] mutant showed a slightly enhanced autophagy flux (*p* value 0.05424) and the Atg13[CD] mutant was 50% defective (Fig. 5B). As observed for the Atg13[44D], also the Atg13[MD] mutant showed reduced Atg17 binding, suggesting that besides S428 and S429, additional phosphorylation sites in the middle region regulate the Atg13-Atg17 interaction. In contrast, the Atg13[CD] mutant co-purified similar amounts of Atg17 as the Atg13[wt] (Fig. 5C). However, all three aspartate mutants showed decreased binding of Atg1 and Atg13 (Fig. 5D). This result suggests that the partial autophagy defect observed in the Atg13[CD] mutant stems from a defect in Atg1 binding but not one in the Atg17 interaction. As the mutated residues in Atg13[CD] are outside of the Atg1-interaction region, C-terminal phosphorylation of Atg13 appears to affect Atg1 binding in trans.

If the autophagy defect observed in the Atg13[CD] mutant only is due to a failure in Atg1 binding, then artificial attachment of Atg1 to Atg13[CD] should rescue the defect. To investigate this possibility, we first tested if an Atg1-Atg13[wt] fusion protein is functional and can rescue the autophagy defect of *atg1Δ atg13Δ* mutant cells.

Indeed, the stable fusion of Atg1 with Atg13[wt] fully restored autophagy in *atg1Δ atg13Δ* cells, similar to the rescue of *atg1Δ* cells by Atg1 expression (Fig. 5E). Next, we tested whether the bulk autophagy defect of the previously described Atg13[FV] mutant (F468A V469A), which is unable to bind Atg1, can be restored by fusing it to Atg1 (Kraft et al, 2012). Indeed, the Atg1-Atg13[FV] fusion protein fully restored bulk autophagy flux in *atg1Δ atg13Δ* cells, confirming that the fusion of Atg1 and Atg13 is functional and can bypass the natural interaction of Atg1 with Atg13 (Fig. 5E). Similarly, fusing the Atg13[CD] mutant to Atg1 was able to restore its bulk autophagy activity, supporting that the Atg13[CD] mutant defect stems from its inability to bind to Atg1 (Fig. 5F).

Previous studies have reported that the initiation of PAS formation is primarily dependent on Atg13 and Atg17, whereas Atg1 kinase activity is required for membrane expansion (Fujioka et al, 2020; Cheong et al, 2008). Therefore, it was anticipated that the Atg13[CD] mutant would hinder autophagy progression rather than impede PAS formation. However, examination of early PAS puncta formation of Atg13[CD]-GFP expressing cells by fluorescent microscopy revealed a significant defect. Remarkably, this defect was rescued by expressing the Atg1-Atg13[CD]-GFP fusion protein (Fig. 5G).

These findings reveal that, in addition to the Atg13-Atg17 interaction, the association of Atg1 and Atg13 plays an essential role in the initial step of PAS formation, highlighting the importance of Atg1 for bulk autophagy initiation.

## Atg11 contributes to PAS formation and bulk autophagy function

The Atg13[MD] and Atg13[44D] mutants showed a strong reduction in the autophagy flux to about 20% and 10% of wild-type levels, respectively. As both these mutants showed reduced Atg1 binding, we tested if their fusion to Atg1 partially rescued this defect. Indeed, the bulk autophagy activity of these mutants was increased to 45% and 35% when stably fused to Atg1 (Fig. 6A). The amount of restored autophagy activity was surprising, given that Atg13[44D] and Atg13[MD] showed hardly any Atg17 binding (Fig. 5C). Therefore, we hypothesized that Atg11 might support bulk autophagy under these conditions, similar to what we observed for Atg13[44A]-induced bulk autophagy under nutrient-rich conditions. To test this possibility, we measured the autophagy flux of the Atg1-Atg13[MD] and Atg1-Atg13[44D] fusion mutants in the absence of Atg11. Indeed, upon deletion of Atg11, we observed a complete autophagy defect for both mutants (Fig. 6B). It should be noted that a reduction in bulk autophagy of about 30% was also observed for the

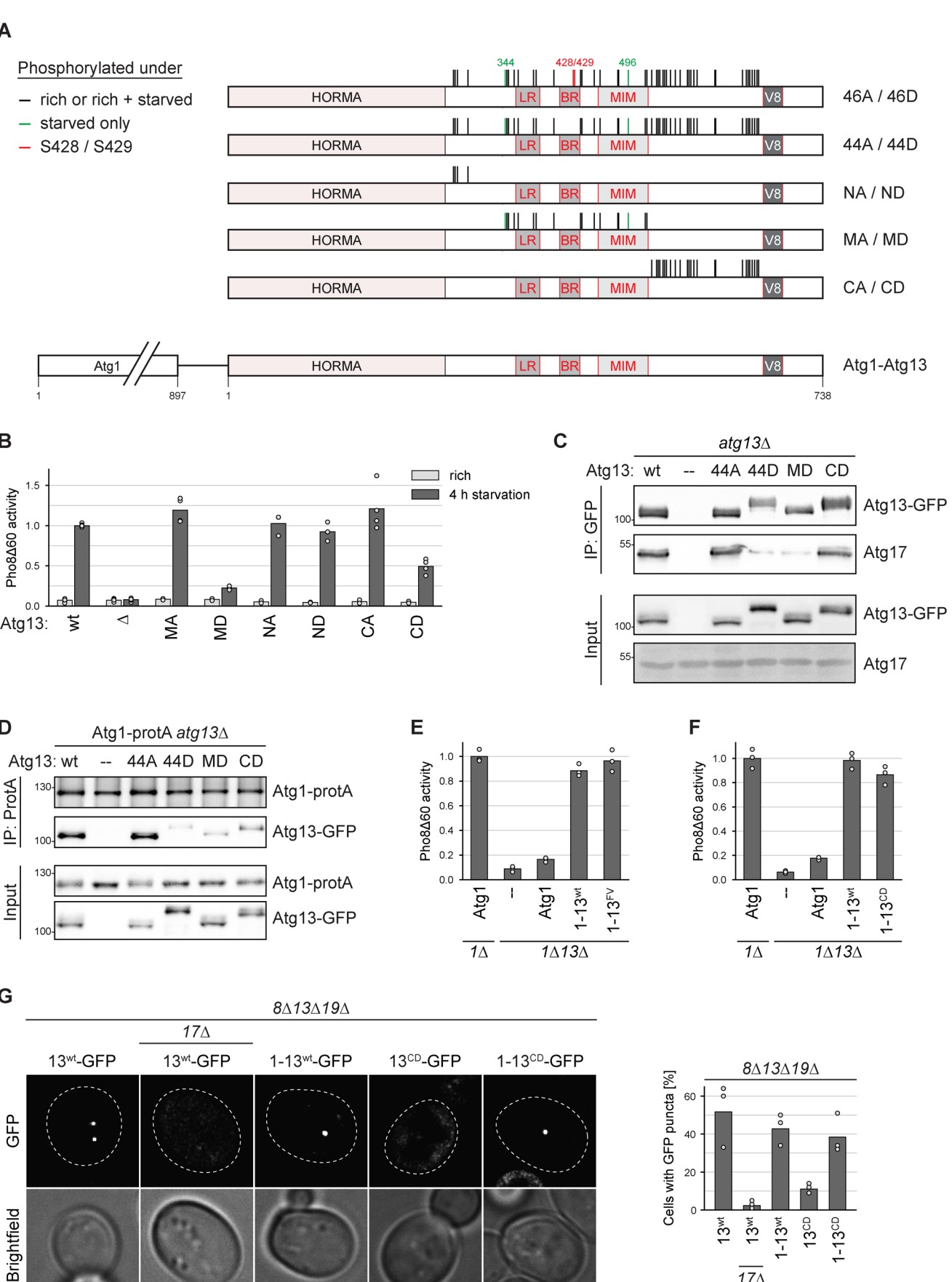

**Figure 5.  Atg13 C-terminal phosphorylations impact Atg1 interaction in trans and control PAS formation.**

(A) Cartoon of Atg13 indicating domain-specific phosphorylation mutants. Black lines indicate phosphorylation sites detected under nutrient-rich conditions only or under nutrient-rich and starvation conditions, red marks S428 and S429, and green depict phosphorylation sites only found under starvation. The latter sites were mutated reciprocally in the respective mutants. (B) Atg13$^{wt}$ pho8$\Delta$60, atg13$\Delta$ pho8$\Delta$60, Atg13$^{MA}$ pho8$\Delta$60, Atg13$^{MD}$ pho8$\Delta$60, Atg13$^{NA}$ pho8$\Delta$60, Atg13$^{ND}$ pho8$\Delta$60, Atg13$^{CA}$ pho8$\Delta$60, and Atg13$^{CD}$ pho8$\Delta$60 cells were grown to log phase and starved for 4 h. Pho8$\Delta$60 alkaline phosphatase activity was measured in three independent experiments. The values of each replicate (circles) and mean (bars) were plotted. All values were normalized to the mean Pho8$\Delta$60 alkaline phosphatase activity of Atg13$^{wt}$ pho8$\Delta$60 cells. (C) Atg1-protA atg13$\Delta$ cells containing plasmid expressed Atg13$^{wt}$-GFP (wt), Atg13$^{44A}$-GFP, Atg13$^{44D}$-GFP, Atg13$^{MD}$-GFP, Atg13$^{CD}$-GFP or an empty plasmid were starved for 1 h. GFP-tagged proteins were immunoprecipitated and the amount of precipitated Atg13 and co-precipitating Atg17 was analyzed by anti-GFP and anti-Atg17 western blotting. One out of three independent biological replicates is shown. (D) Atg1-protA atg13$\Delta$ cells containing plasmid expressed Atg13$^{wt}$-GFP (wt), Atg13$^{44A}$-GFP, Atg13$^{44D}$-GFP, Atg13$^{MD}$-GFP, Atg13$^{CD}$-GFP or an empty plasmid were starved for 1 h. Atg1-protA was immunoprecipitated using IgG beads, and immunoprecipitates were analyzed by anti-Atg1 and anti-Atg13 western blotting. One out of two independent biological replicates is shown. (E) atg1$\Delta$ pho8$\Delta$60 or atg1$\Delta$ atg13$\Delta$ pho8$\Delta$60 cells containing plasmid expressed Atg1, Atg1-13$^{wt}$, Atg1-13$^{FV}$ or an empty plasmid were starved for 4 h. Pho8$\Delta$60 alkaline phosphatase activity was measured in three independent experiments. The values of each replicate (circles) and mean (bars) were plotted. All values were normalized to the mean Pho8$\Delta$60 alkaline phosphatase activity of cells expressing Atg1. The mutants in this experiment were analyzed in parallel with the mutants shown in Fig. 6A. Therefore, the data for the control samples and the Atg1-13$^{wt}$ fusion are the same. (F) atg1$\Delta$ pho8$\Delta$60 or atg1$\Delta$ atg13$\Delta$ pho8$\Delta$60 cells containing plasmid expressed Atg1, Atg1-13$^{wt}$, Atg1-13$^{CD}$ or an empty plasmid were starved for 4 h. Pho8$\Delta$60 alkaline phosphatase activity was measured in three independent experiments. The values of each replicate (circles) and mean (bars) were plotted. All values were normalized to the mean Pho8$\Delta$60 alkaline phosphatase activity of cells expressing Atg1. (G) atg8$\Delta$ atg13$\Delta$ atg19$\Delta$ or atg8$\Delta$ atg13$\Delta$ atg17$\Delta$ atg19$\Delta$ cells containing plasmid expressed Atg13$^{wt}$-GFP, Atg1-13$^{wt}$-GFP, Atg13$^{CD}$-GFP or Atg1-13$^{CD}$-GFP were starved for 30 min and analyzed by fluorescence microscopy. The percentage of cells with GFP puncta was quantified in three independent experiments. For each strain and replicate at least 100 cells were analyzed. Scale bar: 2 µm. Source data are available online for this figure.

Atg1-Atg13$^{wt}$ fusion in the atg11$\Delta$ background. As expected, these mutants also showed an Ape1 processing defect in the absence of Atg11 or Atg19 (Fig. 6C). These findings substantiate an active involvement of Atg11 in bulk autophagy and suggest that a synthetic defect arises from the impaired Atg13-Atg17 interaction combined with the loss of Atg11.

To test if Atg11 is indeed involved in a similar process as the Atg13-Atg17 interaction, we used fluorescence microscopy to assess the impact of Atg11 on the initial formation of the PAS. While some PAS puncta were formed by Atg1-Atg13$^{MD}$-GFP and Atg1-Atg13$^{44D}$-GFP in the presence of Atg11, almost no puncta were observed in atg11$\Delta$ cells (Fig. 6D). These findings suggest that Atg11 operates in an analogous manner to the Atg13-Atg17 interaction.

### The Atg11-dependent bulk PAS possesses similar characteristics as the wild-type PAS

The formation of the early bulk autophagy PAS has been reported to involve the creation of a liquid-like condensate, facilitated by multivalent interaction between Atg13 and Atg17 (Fujioka et al, 2020). Given the requirement of Atg11 for the formation of the early bulk autophagy PAS in the Atg1-Atg13$^{44D}$ and Atg1-Atg13$^{MD}$ fusion mutants, we investigated whether this PAS structure exhibits comparable liquid-like characteristics to those observed in wild-type conditions. To assess this, we utilized 1,6-hexanediol, a widely employed chemical for disrupting liquid–liquid phase separated assemblies, which has also been shown to affect the bulk autophagy PAS in yeast cells (Fujioka et al, 2020). As expected, 1,6-hexanediol effectively dissolved wild-type PAS puncta, which were monitored by expressing Atg13-GFP in atg13$\Delta$ atg19$\Delta$ cells. Similarly, GFP puncta formed in Atg1-Atg13$^{MD}$-GFP and Atg1-Atg13$^{44D}$-GFP containing atg1$\Delta$ atg13$\Delta$ atg19$\Delta$ cells vanished upon 1,6-hexanediol treatment, indicating that the Atg11-dependent bulk autophagy PAS also forms through liquid–liquid phase separation and that Atg11 somehow contributes to this process (Fig. 6E).

This study demonstrates that the phosphorylation of numerous serine and threonine residues on Atg13 can modulate its interaction with Atg1 and Atg17 and that both of these interactions

are crucial for the initiation of bulk PAS formation. Furthermore, our findings uncover an intricate molecular assembly process of the PAS, involving not only the multivalent interaction between Atg13 and Atg17 but also the participation of Atg1 and Atg11 (Fig. 7).

## Discussion

In this study, we took a comprehensive approach to investigate the function of the numerous phosphorylation events that occur on Atg13 during nutrient-rich conditions or nitrogen starvation. Our findings demonstrate that the post-translational regulation of Atg13 by numerous phosphorylation events is a crucial mechanism for precisely regulating bulk autophagy activity.

Different studies have reported many in vivo phosphorylation sites on Atg13, but only a few of these sites have been functionally characterized. We mapped 48 in vivo phosphorylation sites, several of them overlapping with previous studies. To test whether mutation of such a large number of phosphorylation sites is not simply disrupting the function of Atg13, we generated two reciprocal alleles by mutating the same set of phospho-accepting residues to either alanine or aspartate. Atg13$^{44A}$ mimics the dephosphorylated, active state, and Atg13$^{44D}$ mimics the phosphorylated, inactive state. Despite the large number of mutated residues, both mutants were stably expressed. Indeed, the Atg13$^{44D}$ mutant was fully defective in bulk autophagy, whereas Atg13$^{44A}$ was not only fully functional but displayed strong hyperactivity. Atg13$^{44A}$ was even able to induce bulk autophagy under nutrient-rich conditions when TORC1 is active. To our knowledge, this is the first pair of reciprocal mutants that can mimic the fully active and the fully inhibited state of Atg13. Importantly, after long-term starvation, the hyperactive Atg13$^{44A}$ mutant showed decreased survival, a phenotype that had so far exclusively been observed for autophagy-defective mutants. These findings demonstrate that disrupting the dynamic regulation of Atg13 can lead to insufficient or overshooting autophagy, both of which are detrimental to cell survival.

Phosphorylations occurring in different regions of Atg13 play a crucial role in regulating its interactions with Atg1 and Atg17

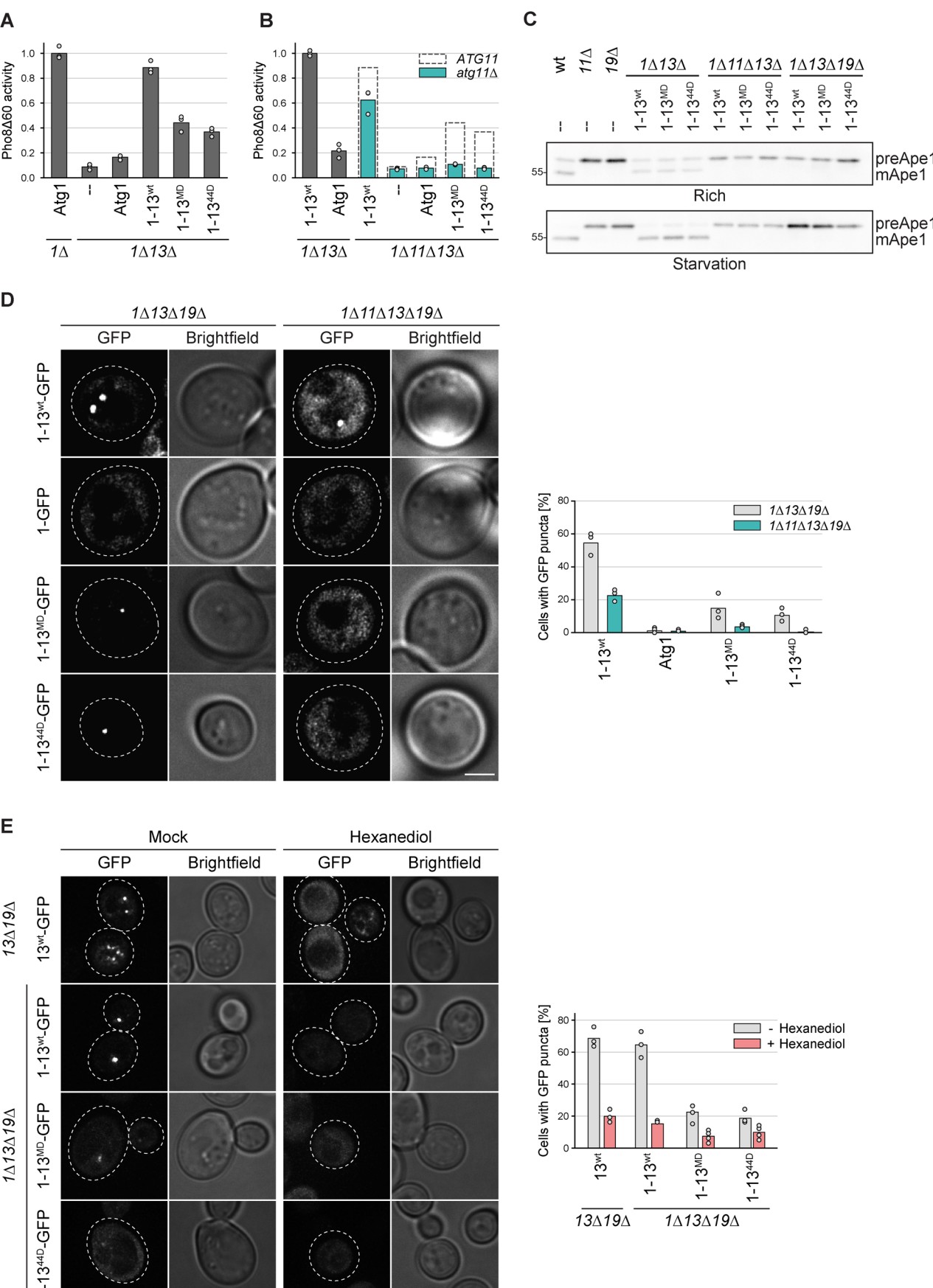

**Figure 6. Atg11 contributes to PAS formation and bulk autophagy function.**

(A) atg1Δ pho8Δ60 or atg1Δ atg13Δ pho8Δ60 cells containing plasmid expressed Atg1, Atg1-13$^{wt}$, Atg1-13$^{MD}$, Atg1-13$^{44D}$ or an empty plasmid were starved for 4 h. Pho8Δ60 alkaline phosphatase activity was measured in three independent experiments. The values of each replicate (circles) and mean (bars) were plotted. All values were normalized to the mean Pho8Δ60 alkaline phosphatase activity of cells expressing Atg1. The mutants in this experiment were analyzed in parallel with the mutants shown in Fig. 5E. Therefore, the data for the control samples and the Atg1-13$^{wt}$ fusion are the same. (B) atg1Δ atg13Δ pho8Δ60 or atg1Δ atg11Δ atg13Δ pho8Δ60 cells containing plasmid expressed Atg1, Atg1-13$^{wt}$, Atg1-13$^{MD}$, Atg1-13$^{44D}$ or an empty plasmid were starved for 4 h. Pho8Δ60 alkaline phosphatase activity was measured in three independent experiments. The values of each replicate (circles) and mean (bars) were plotted. All values were normalized to the mean Pho8Δ60 alkaline phosphatase activity of cells expressing Atg1-13$^{wt}$. Dashed rectangles represent the mean Pho8Δ60 alkaline phosphatase activity of the respective mutant measured in atg1Δ ATG11 atg13Δ pho8Δ60 cells shown in panel (A). (C) Wild-type (wt), atg11Δ and atg19Δ cells containing an empty plasmid and atg1Δ atg13Δ, atg1Δ atg11Δ atg13Δ, and atg1Δ atg13Δ atg19Δ cells containing plasmid expressed Atg1-13$^{wt}$, Atg1-13$^{MD}$, and Atg1-13$^{44D}$ were grown to log phase and starved for 4 h, and cell extracts were prepared by TCA precipitation. Ape 1 processing was analyzed by anti-Ape1 western blotting. One out of two independent biological replicates is shown. (D) atg1Δ atg13Δ atg19Δ or atg1Δ atg11Δ atg13Δ atg19Δ cells containing plasmid expressed Atg1-13$^{wt}$-GFP, Atg1-GFP, Atg1-13$^{MD}$-GFP, or Atg1-13$^{44D}$-GFP were starved for 30 min. The percentage of cells with GFP puncta was quantified in three independent experiments. For each strain and replicate at least 100 cells were analyzed. Scale bar: 2 μm. (E) atg13Δ atg19Δ or atg1Δ atg13Δ atg19Δ cells containing plasmid expressed Atg13$^{wt}$-GFP, Atg1-13$^{wt}$-GFP, Atg1-13$^{MD}$-GFP or Atg1-13$^{44D}$-GFP were starved for 30 min and treated with 1,6-hexanediol or mock treated with digitonin for 3 min, and analyzed by fluorescence microscopy. The percentage of cells with GFP puncta was quantified in three or four independent experiments. For each strain and replicate at least 100 cells were analyzed. Scale bar: 2 μm. Source data are available online for this figure.

through different mechanisms. The central region of Atg13 contains the Atg1 and Atg17 binding domains. We identified previously uncharacterized phosphorylation sites within these domains that regulate the interaction of Atg13 with Atg1 and Atg17. Moreover, phosphorylation in the C-terminal region of Atg13, which is not required for Atg1 binding per se, negatively affects the interaction with Atg1 in trans. These additional levels of regulation may contribute to the intricate mechanism that controls the activity of the bulk autophagy process and facilitates the integration of various cellular signals on Atg13. Whether all of the mutated residues in the middle and C-terminal region of Atg13 contribute to the regulation of bulk autophagy or if this function is governed only by a subset of the sites needs further investigation. Given that an Atg1-Atg13 fusion protein seems completely operational in bulk autophagy upon nitrogen starvation, the question arises if a phospho-regulated detachment of the Atg1-Atg13 complex, post PAS formation, happens at all or if such a detachment has any functional consequences during autophagosome generation.

Initially, Atg11 was thought to play a role only in selective autophagy pathways, but more recent work has demonstrated its involvement in bulk autophagy as well. This involvement has primarily been observed in conditions of bulk autophagy triggered by means other than nitrogen starvation, such as carbon or phosphate starvation, (Adachi et al, 2017; Yao et al, 2020; Yokota et al, 2017). Nonetheless, a possible involvement of Atg11 in nitrogen-starvation-induced bulk autophagy has been occasionally observed (Liu et al, 2016; Mao et al, 2013). Notably, fission yeast appears to not only rely on Atg17 but also on Atg11 for its bulk autophagy function (Pan et al, 2020). In line with a role of Atg11 also in bulk autophagy of *S. cerevisiae*, we observed that the previously reported Atg17-binding mutant Atg13$^{S379D}$ (Yamamoto et al, 2016) only marginally affected bulk autophagy under nitrogen starvation in the presence of Atg11, however, showed an enhanced defect in *atg11Δ* mutants (Fig. 1D). Moreover, the bulk autophagy activity induced by Atg13$^{44A}$ under nutrient-rich conditions was severely reduced in the absence of Atg11 (Fig. 4A). Similarly, both the Atg1-Atg13$^{MD}$ and Atg1-Atg13$^{44D}$ fusion proteins, which almost completely lacked Atg17 binding (Fig. 5B), still showed up to 45% bulk autophagy flux upon starvation but were completely defective in the absence of Atg11 (Fig. 6A,B). This unexpected synthetic defect in bulk autophagy, caused by the deletion of Atg11 in

combination with a weakened Atg13-Atg17 interaction, resulted in a complete failure in PAS formation (Fig. 6D), strongly supporting a direct involvement of Atg11 in early PAS generation also in bulk autophagy. All these observations hint at an involvement of Atg11 in bulk autophagy, also during nitrogen starvation. Different starvation conditions have been reported to induce varying levels of activity in bulk autophagy (Yokota et al, 2017; Rangarajan et al, 2020). This variability in activity may be attributed to variations in the extent of Atg13 dephosphorylation, which in turn affects its affinity for Atg1 and Atg17. Our results suggest that when the affinity of Atg13 for Atg17 is only partially increased, the importance of Atg11 in PAS assembly becomes more pronounced, which might explain the observed dependency on Atg11 under certain starvation conditions. Comparing the phosphorylation status of Atg13 across different starvation conditions would provide valuable insights into whether the degree of dephosphorylation correlates with the level of bulk autophagy flux.

Phase separation has been reported to play a critical role in autophagy by organizing the autophagy machinery at the PAS (Fujioka et al, 2020). It was suggested that the interaction between Atg13 and Atg17 drives the phase transition of the PAS and that the phosphorylation status of Atg13 functions as a regulator for this process. Phase separation is typically driven by multivalent interactions of relatively low strength. Previous studies have established a direct interaction of Atg11 with Atg1 and Atg29, and that Atg11 exists as a dimeric protein (Mao et al, 2013; Suzuki and Noda, 2018). Based on our findings, it is plausible to propose that the weak interactions between Atg1, Atg11, and Atg29 contribute to the pool of multivalent interactions among the early PAS proteins that drive the phase transition of the PAS. This model raises intriguing questions about the mechanisms of autophagy induction in other organisms that possess a different set of autophagy-related proteins. It is possible that the phase separation of the PAS may involve varying sets of proteins, while they have in common to rely on a pool of multivalent interactions. This flexibility in utilizing different sets of proteins would allow autophagy initiation to be controlled by various signaling pathways that regulate the interaction capabilities of core and auxiliary autophagy proteins. Additional research is required to unravel the contribution of different early PAS factors in *S. cerevisiae* to phase separation at a molecular level, as well as how this process is achieved in other organisms with distinct autophagy factors.

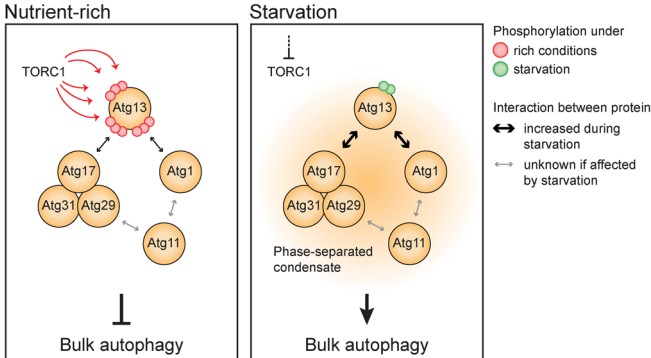

**Figure 7. Atg13 acts as a signaling hub, controlling the formation of the phase-separated PAS in bulk autophagy.**

Under nutrient-rich conditions (left panel), TORC1-dependent phosphorylation prevents the tight association of Atg13 with Atg17 and Atg1, preventing the initiation of bulk autophagy. When starvation occurs (right panel), TORC1 activity is inhibited, leading to the dephosphorylation of Atg13. Dephosphorylated Atg13 exhibits increased affinity for Atg17 and Atg1, serving as a switch that triggers the phase separation of the PAS. The phase-separated condensate is formed by numerous multivalent interactions involving Atg13, the Atg17 complex, Atg1, and Atg11. Consequently, Atg13 acts as a signaling hub, with its phosphorylation status regulating the level of bulk autophagy activity and allowing precise control over the process.

# Methods

## Yeast strains and plasmids

Yeast strains are listed in Table 1. Plasmids are listed in Tables 2, 3. Yeast genomic mutations were integrated by homologous recombination, and multiple deletions or mutations were generated by PCR knockout, mating, and dissection.

## Growth condition

Yeast grown in a synthetic medium (SD: 0.17% yeast nitrogen base, 0.5% ammonium sulfate, 2% glucose, and amino acids) or YPD (1% yeast extract, 2% peptone, and 2% glucose). For starvation, cells were washed with and resuspended in a nitrogen starvation medium (SD-N: 0.17% yeast nitrogen base without amino acids, 2% glucose) for the indicated time to induce bulk autophagy. Yeast cultures were incubated at 30 °C with shaking at 220 rpm. For hydroxyurea (HU) treatment, yeast cultures were grown overnight in YPD or SD medium in logarithmic (log) phase and diluted to an $OD_{600}$ of 0.3. 2 M HU stock solution was freshly prepared in YPD or synthetic medium and added to a final concentration of 200 mM. Cells were further incubated for 4–8 h.

## Antibodies

Antisera were diluted in 1x PBS pH 7.4 containing 5% milk powder: Mouse monoclonal anti-GFP (1:100, clone 2B6, Merck, cat # MABC1689), rabbit polyclonal PAP antibody (1:3000, Sigma-Aldrich, cat # P1291), mouse monoclonal anti-Pgk1 antibody (1:10,000, clone 22C5D8, Invitrogen, cat # 459250), rabbit polyclonal anti-Atg1 antibody (1:15,000, Daniel Klionsky, University of Michigan, USA), rabbit polyclonal anti-Atg13 antibody (1:15,000, Daniel Klionsky, University of Michigan, USA), mouse monoclonal anti-Atg11 antibody (1:500, clone 6F4-G4, Claudine Kraft, University of Freiburg, Germany), mouse monoclonal anti-Atg17 antibody (1:50, clone 4D3-E8, Claudine Kraft, University of Freiburg, Germany), mouse monoclonal anti-Myc antibody (used 1:5000) (4A6, Millipore). Rabbit polyclonal anti-pT737- Sch9 (used 1:500, Robbie Loewith, University of Geneva, Switzerland) was diluted in 1× PBS pH 7.4 containing 5% BSA.

## Pho8Δ60 assay

15-20 $OD_{600}$ units of yeast culture were harvested by centrifugation ($2000 \times g$, 5 min, room temperature). Pellets were washed in 1 ml distilled water, followed by centrifugation ($2000 \times g$, 5 min, 4 °C). Pellets were resuspended in 2 ml ice-cold 0.85% NaCl containing 1 mM PMSF, centrifuged ($2000 \times g$, 5 min, 4 °C), and snap-frozen in liquid nitrogen. Pellets were resuspended in 16 μl/$OD_{600}$ unit lysis buffer (20 mM PIPES pH 6.8, 0.5% Triton X-100, 50 mM NaCl, 100 mM potassium acetate, 10 mM $MgSO_4$, 10 μM $ZnSO_4$, 1 mM PMSF, cOmplete™ protease inhibitor cocktail (EDTA-free, Roche)). Cells were lysed by bead beating for 5 min at 4 °C and cell extracts were cleared by centrifugation ($16,000 \times g$, 5 min, 4 °C). The supernatant was diluted with lysis buffer to a protein concentration of 50 μg/100 μl. 400 μl reaction buffer (0.4% Triton X-100, 10 mM $MgSO_4$, 10 μM $ZnSO_4$, and 250 mM Tris-HCl pH 8.5) containing 6.25 mM alpha-naphthyl phosphate (Sigma-Aldrich) was added to the enzymatic reactions and to the control reaction, which only contained lysis buffer. Reactions were incubated at 37 °C for 10 min and stopped by adding 500 μl stop buffer (1 M glycine pH 11). Fluorescence was measured using 345 nm for excitation and 472 nm for emission. For each experiment, at least three independent replicates were performed. To calculate Normalized Pho8Δ60 activity, for each batch of samples a standard curve was generated using a dilution series of 1-naphthol (Sigma-Aldrich). A second-degree polynomial curve was fitted to the standard curve, which was then used for the calculation of relative activity values for each sample. If any data points exceeded the highest value of the standard curve, a linear interpolation was applied using the slope from the highest point of the polynomial curve. To ensure comparability between different batches, a normalization procedure was implemented. Firstly, samples that were measured in all batches were selected. A normalization factor was calculated as the average relative activity of these samples. Within each batch, the relative activity of all samples was divided by the respective normalization factor. Finally, the relative activity of all samples was divided by the mean relative activity of the reference condition, as indicated in the figure legends, thereby setting the Pho8Δ60 activity of the reference condition to 1. Statistical analysis was performed using two-tailed unpaired $t$-tests.

## Serial dilution spot assay

Yeast cell cultures were grown to mid-log phase in YPD medium, shifted to SD-N medium containing 10 μg/ml tetracycline, diluted to an $OD_{600}$ of 0.2, and directly spotted or incubated at 30 °C for 14 days prior to spotting. For analyzing cell viability, a 7-fold serial dilution series was prepared by diluting cell cultures in SD-N medium. 4 μl of each dilution was spotted onto YPD plates and incubated at 30 °C. Pictures were taken after 32 h of incubation at 30 °C.

**Table 1. Yeast strains used in this study.**

| Name | Genotype | Background | Source |
|---|---|---|---|
| BY4741 | his3Δ1 leu2Δ0 met15Δ0 ura3Δ0; Mat a | BY474x | Euroscarf |
| yABH1 | pho8::pho8Δ60:His ATG13$^{wt}$:Ura; Mat alpha | BY474x | This study |
| yABH2 | pho8::pho8Δ60:His ATG13$^{CD}$:Ura; Mat alpha | BY474x | This study |
| yABH3 | pho8::pho8Δ60:His ATG13$^{44D}$:Ura; Mat alpha | BY474x | This study |
| yABH6 | pho8::pho8Δ60:His ATG13$^{NA}$:Ura; Mat alpha | BY474x | This study |
| yABH7 | pho8::pho8Δ60:His ATG13$^{ND}$:Ura; Mat alpha | BY474x | This study |
| yABH8 | pho8::pho8Δ60:His ATG13$^{CA}$:Ura; Mat alpha | BY474x | This study |
| yABH11 | ATG13$^{wt}$:Ura; Mat a | BY474x | This study |
| yABH17 | pho8::pho8Δ60:His ATG13$^{44A}$:Ura; Mat alpha | BY474x | This study |
| yABH25 | ATG13$^{44D}$:Ura; Mat a | BY474x | This study |
| yABH26 | ATG13$^{44A}$:Ura; Mat a | BY474x | This study |
| yABH59 | pho8::pho8Δ60:His ATG13$^{8SA}$:Ura; Mat alpha | BY474x | This study |
| yABH60 | pho8::pho8Δ60:His ATG13$^{8SD}$:Ura; Mat alpha | BY474x | This study |
| yABH62 | pho8::pho8Δ60:His ATG13$^{MA}$:Ura; Mat alpha | BY474x | This study |
| yABH63 | pho8::pho8Δ60:His ATG13$^{MD}$:Ura; Mat alpha | BY474x | This study |
| yABH73 | pho8::pho8Δ60:His atg1::Kan atg11::Hyg atg13::Kan; Mat a | BY474x | This study |
| yABH76 | pho8::pho8Δ60:His ATG13$^{379D}$:Ura; Mat alpha | BY474x | This study |
| yABH78 | pho8::pho8Δ60:His ATG13$^{428D429D}$:Ura; Mat alpha | BY474x | This study |
| yABH79 | pho8::pho8Δ60:His ATG13$^{429D}$:Ura; Mat alpha | BY474x | This study |
| yABH80 | pho8::pho8Δ60:His ATG13$^{428A429A}$:Ura; Mat alpha | BY474x | This study |
| yABH81 | pho8::pho8Δ60:His atg11::Nat ATG13$^{379D}$:Ura; Mat a | BY474x | This study |
| yABH86 | pho8::pho8Δ60:His atg11::Nat ATG13$^{wt}$:Ura; Mat a | BY474x | This study |
| yABH90 | atg8::His atg13::Kan atg19::Hyg; Mat a | BY474x | This study |
| yABH91 | atg8::Kan atg13::Nat atg17::His atg19::Kan; Mat a | BY474x | This study |
| yCK416 | atg11:Kan; Mat a | BY474x | (Pfaffenwimmer et al, 2014) |
| yCK418 | atg13::Kan; Mat a | BY474x | This study |
| yDH11 | pho8::pho8Δ60:His; Mat alpha | BY474x | This study |
| yDH516 | pho8::pho8Δ60:His atg13::Kan atg1::Kan; Mat alpha | BY474x | This study |
| yDP441 | atg1::Kan atg13::Nat atg19::Kan; Mat a | BY474x | This study |
| yDP445 | atg13::Nat atg19::Kan; Mat a | BY474x | This study |
| yDP451 | atg19::Hyg; Mat a | BY474x | This study |
| yDP454 | atg1::Kan atg11::Hyg atg13::Nat atg19::Kan; Mat a | BY474x | This study |
| yRT15 | Atg1-stag(protA-speptide):His atg13::Nat atg17-13myc:Kan Vac8-HA:Kan; Mat a | BY474x | This study |
| yRT70 | atg1::Kan atg11::Hyg atg13::Kan; Mat alpha | BY474x | (Torggler et al, 2016) |
| yRT105 | atg1::His atg13::Nat; Mat alpha | BY474x | This study |
| yRT199 | pho8::pho8Δ60:His atg1::Kan; Mat a | BY474x | This study |
| yRT238 | atg13::Kan pep4Δ::Nat; Mat a | BY474x | This study |
| ySS15 | pho8::pho8Δ60:His atg13::Kan; Mat alpha | BY474x | This study |
| ySS22 | pho8::pho8Δ60:His atg13::Kan atg11::Nat; Mat a | BY474x | This study |
| ySS24 | pho8::pho8Δ60:His atg13::Kan atg17::His; Mat a | BY474x | This study |
| ySS31 | pho8::pho8Δ60:His atg13::Kan atg19::Kan; Mat alpha | BY474x | This study |
| yTP210 | Atg1-stag(protA-speptide):His atg13::Nat; Mat alpha | BY474x | (Torggler et al, 2016) |

Description: Atg13 constructs were genomically integrated by homologous recombination into the atg13::Kan locus in the following BY474x strains: pho8::pho8Δ60:His atg13::Kan (ySS15) or atg13::Kan (yCK418).

**Table 2. Centromeric plasmids used in this study.**

| Name | Characteristics | Promoter | Terminator | Source |
|---|---|---|---|---|
| pRS315 | CEN, LEU2 | – | – | (Sikorski and Hieter, 1989) |
| pRS316 | CEN, URA3 | – | – | (Sikorski and Hieter, 1989) |
| pABH15 | Atg1-13$^{CD}$; pRS415 | Atg1 | CYC1 | This study |
| pABH41 | Atg13$^{8SD}$; pRS316 | Atg13 | Atg13 | This study |
| pABH45 | Atg1-13$^{MD}$; pRS415 | Atg1 | CYC1 | This study |
| pABH46 | Atg1-13$^{44D}$; pRS415 | Atg1 | CYC1 | This study |
| pABH48 | Atg13$^{44A}$-GFP; pRS415 | Atg13 | CYC1 | This study |
| pABH49 | Atg13$^{44D}$-GFP; pRS415 | Atg13 | CYC1 | This study |
| pABH53 | Atg13$^{MD}$-GFP; pRS415 | Atg13 | CYC1 | This study |
| pABH55 | Atg13$^{CD}$-GFP; pRS415 | Atg13 | CYC1 | This study |
| pABH65 | Atg1-13$^{CD}$-GFP; pRS415 | Atg1 | CYC1 | This study |
| pABH66 | Atg1-13$^{wt}$-GFP; pRS415 | Atg1 | CYC1 | This study |
| pABH71 | Atg13$^{7SD1A}$; pRS316 | Atg13 | Atg13 | This study |
| pCK111 | Atg1; pRS315 | Atg1 | Atg1 | (Kijanska et al, 2010) |
| pCK582 | Atg13$^{wt}$-GFP; pRS415 | Atg13 | CYC1 | (Torggler et al, 2016) |
| pCK789 | Atg13; pRS316 | Atg13 | Atg13 | This study |
| pDP227 | Atg1-GFP; pRS415 | Atg1 | CYC1 | This study |
| pMS98 | Atg1-13$^{FV}$; pRS415 | Atg1 | CYC1 | This study |
| pRT09 | Atg13$^{46A}$; pRS316 | Atg13 | Atg13 | This study |
| pRT10 | Atg13$^{46D}$; pRS316 | Atg13 | Atg13 | This study |
| pRT15 | Atg13$^{44A}$; pRS316 | Atg13 | Atg13 | This study |
| pRT16 | Atg13$^{44D}$; pRS316 | Atg13 | Atg13 | This study |
| pRT67 | Atg1-13$^{wt}$; pRS415 | Atg1 | CYC1 | This study |

**Table 3. Integrative plasmids used to generate strains.**

| Name | Characteristics | Promoter | Terminator | Source |
|---|---|---|---|---|
| pABH03 | Atg13$^{wt}$, pRS406 | Atg13 | Atg13 | This study |
| pABH04 | Atg13$^{CD}$, pRS406 | Atg13 | Atg13 | This study |
| pABH05 | Atg13$^{44D}$, pRS406 | Atg13 | Atg13 | This study |
| pABH08 | Atg13$^{NA}$, pRS406 | Atg13 | Atg13 | This study |
| pABH09 | Atg13$^{ND}$, pRS406 | Atg13 | Atg13 | This study |
| pABH10 | Atg13$^{CA}$, pRS406 | Atg13 | Atg13 | This study |
| pABH13 | Atg13$^{44A}$, pRS406 | Atg13 | Atg13 | This study |
| pABH34 | Atg13$^{MA}$, pRS406 | Atg13 | Atg13 | This study |
| pABH35 | Atg13$^{MD}$, pRS406 | Atg13 | Atg13 | This study |
| pABH42 | Atg13$^{8SA}$, pRS406 | Atg13 | Atg13 | This study |
| pABH43 | Atg13$^{8SD}$, pRS406 | Atg13 | Atg13 | This study |
| pABH57 | Atg13$^{379D}$, pRS406 | Atg13 | Atg13 | This study |
| pABH58 | Atg13$^{429D}$, pRS406 | Atg13 | Atg13 | This study |
| pABH60 | Atg13$^{428D429D}$, pRS406 | Atg13 | Atg13 | This study |
| pABH61 | Atg13$^{428A429A}$, pRS406 | Atg13 | Atg13 | This study |

## Pulldown assays

0.75 g of freezer-milled yeast powder was thawed on ice and 468 µl of IP buffer was added. The extract was cleared by centrifugation (twice for 10 min, $5000 \times g$, 4 °C). Protein concentration was adjusted to 20 µg/µl in 600–700 µl IP buffer. For protein A pulldowns, the extract was incubated with 5 µl of Dynabeads™ M-270 Epoxy (Invitrogen), which were coupled with IgG from rabbit serum (Sigma). For GFP pulldowns, 5 µl of GFP-Trap® Dynabeads™ (Chromo Tek) were used. Beads with extract were rotated for 1 h at 4 °C. Beads were washed three times for 5 min in IP buffer and finally eluted in 10 µl of urea loading buffer.

## Electron microscopy

15 OD$_{600}$ units of cells were harvested by centrifugation ($1800 \times g$, 5 min, room temperature). Cells were washed in distilled water and pelleted by centrifugation ($1800 \times g$, 5 min, room temperature). Cells were resuspended in 3 ml of freshly prepared ice-cold 1.5% KMnO$_4$ (Sigma) and transferred into two 1.5 ml microfuge tubes. After topping up the tube with the same solution to exclude air, samples were mixed for 30 min rotating at 4 °C. After centrifugation ($1400 \times g$, 3 min, 4 °C), the 1.5% KMnO$_4$ incubation was repeated once more before washing the pellets five times with 1 ml of distilled water. Permanganate-fixed cells were dehydrated stepwise with increasing concentrations of acetone (10%, 30%, 50%, 70%, 90%, 95%, and three times 100%-dried acetone). Each incubation step was performed for 20 min at at room temperature on a slow motion (4 rpm) rotatory wheel, and in between each step cells were pelleted by centrifugation ($1400 \times g$, 3 min, room temperature). The 100% Spurr's resin was freshly prepared by mixing 10 g of 4-vinylcyclohexene dioxide (or ERL4206) (Sigma), 4 g of epichlorohydrin-polyglycol epoxy (DER) resin 736 (Sigma), 26 g of (2-nonen-1-yl) succinic anhydride (NSA) (Sigma), and 0.4 g of N,N-diethylethanolamine (Ted-Pella). After dehydration, cell pellets were resuspended in 33% Spurr's resin (5 ml of 100% Spurr's

## Yeast extract preparation

For preparing trichloroacetic acid (TCA) extracts, 1.5 ml of yeast cell culture was precipitated with 7% TCA and incubated for 10–20 min on ice. Precipitated proteins were pelleted at $16,000 \times g$ for 5 min at 4 °C, washed with 1 ml acetone, pelleted at $16,000 \times g$ for 8 min at 4 °C, air-dried, resuspended in urea loading buffer (120 mM Tris-HCl pH 6.8, 5% glycerol, 8 M urea, 143 mM beta-mercaptoethanol, 8% SDS), and boiled before SDS-PAGE and western blot analysis. For the preparation of freezer-milled yeast powder, cells were harvested by centrifugation ($3000 \times g$, 10 min, room temperature), washed with 1× PBS containing 2% glucose, and resuspended in 1 µl/OD$_{600}$ unit IP buffer (1× PBS, 10% glycerol, 0.5% Tween-20, 1 mM NaF, 1 mM PMSF, 20 mM β-glycerophosphate, 1 mM Na$_3$VO$_4$ and cOmplete™ protease inhibitor cocktail (Roche)). Cells were milled in a cryogenic grinder (SPEX Freezer Mill 6875, SPEX SamplePrep), using five rounds of 3 min breakage at 15 cycles per second and 2 min cooling. Yeast powder was stored at −80 °C.

resin mixed with 10 ml of acetone dried (MERCK-1.00299.0500)) and incubated on a rotating wheel for 1 h rotating at room temperature. Cells were pelleted (7600 × g, 3 min, room temperature) and incubated in 100% Spurr's resin overnight rotating at room temperature. This procedure was repeated over the day, after centrifugation of the overnight incubation (9000 × g, 5 min, room temperature). Afterward, the cells resuspended in 100% Spurr's resin were transferred to size 00 embedding capsules (Electron Microscopy Science) and were pelleted by centrifugation (9000 × g, 5 min, room temperature). Embedding capsules were topped up with 100% Spurr's resin and baked for a minimum of 3 days at 60 °C. Thin sections of ~55 nm were cut using an UC-7 ultramicrotome (Leica Microsystems). Cell sections were collected on formvar carbon-coated 50 meshes copper grids (EMS) and stained with a filtered lead-citrate solution (80 mM lead nitrate, 120 mM sodium citrate pH 12) for 2 min at room temperature. Sections were viewed in a CM100bio TEM (Thermo-Fischer, Eindhoven).

## Quantitative live-cell imaging

Yeast cells were attached to either 35 mm glass bottom dishes (D35-20, 1.5, In Vitro Scientific) (Figs. 2G, 5G, and 6D) or to Ibidi slides (µ-Slide VI 0.4 ibiTreat) (Fig. 6E) pretreated with concanavalin A type IV (1 mg/ml). Imaging was performed at room temperature. For 1,6-hexanediol treatment, yeast cells were incubated in nitrogen starvation media supplemented with 10 µg/ml digitonin and 10% 1,6-hexanediol. Mock treatment was performed by adding 10 µg/ml digitonin (Kroschwald et al, 2017). Fluorescence microscopy images were recorded with a DeltaVision Ultra High-Resolution microscope (GE Healthcare, Applied Precision) equipped with an UPlanSApo 100x/1.4 oil objective (Olympus), a sCMOS pco.edge camera (PCO), and a seven-channel solid state light source (Lumencor). Raw microscopy images were deconvolved using the softWoRx deconvolution plugin (version 7.2.1). Image analysis was performed using FIJI (Schindelin et al, 2012). Images from each figure panel were taken with the same imaging setup and are shown with the same contrast settings unless stated otherwise. All images were generated by collecting a z-stack of 21 pictures with focal planes 0.25 µm apart. Single focal planes of representative images are shown in Figs. 2G, 5G, and 6D. Collapsed Z-stack focal planes of representative images are shown in Fig. 6E. For quantification independent replicates were analyzed and manual counting was performed blindly after randomizing image names.

## HB purification and mass spectrometry (MS) analysis

Stable isotope labeling using amino acids in cell culture (SILAC) (Ong et al, 2002) was achieved as described previously (Reiter et al, 2012). Histidin-biotin (HB) (Reiter et al, 2012) tandem affinity tag fusions of Atg13 were expressed in S.cerevisiae wild-type and atg1Δ cells. Cells were grown to mid-log phase and starved in SD-N or treated with rapamycin for 45 min, harvested by filtration, and deep-frozen in liquid $N_2$. HB tandem affinity purifications, in-solution digestion with trypsin, and enrichment of phosphorylated peptides using $TiO_2$ was performed as described previously (Reiter et al, 2012). Trypsinized peptides were analyzed on a reversed-phase nano-high-performance LC-MS system (Ultimate 3000 RSLC nano-flow chromatography system, Thermo Fisher Scientific) coupled with an electrospray ionization interface (Proxeon Biosystems). Peptides were separated by applying an increasing organic solvent (acetonitrile) gradient from 2.5% to 40% in 0.1% formic acid (running time 80 min). The capillary temperature was set to 275 °C. MS analysis was performed using a Linear Trap Quadrupole Orbitrap Velos (Thermo Fisher Scientific). The mass spectrometer was operated in data-dependent mode, with a mass range of 350–2000 m/z with lock mass activated, at a resolution of 60'000 at 200 m/z and an automatic gain control (AGC) target value of 3E6. The 12 most intense ions were selected for fragmentation in the collision-induced dissociation mode.

Data analysis was performed using the SEQUEST algorithm (Proteome Discoverer 1.3) using the Saccharomyces Genome Database (SGD, version February 2011) along with contaminants derived from the common laboratory contaminants database (MQ). Fixed modifications included carbamidomethylation of cysteine. Protein N-terminal acetylation, deamidation, oxidation of methionine, phosphorylation of serine, threonine and tyrosine, and heavy labels of arginine and lysine (Arg6, Lys6) were set as variable. Enzyme specificity was set to "Trypsin/P" and a maximum of 2 missed cleavages per peptide was allowed. For the assignment of phosphorylation sites, we integrated the tool phosphoRS into the Proteome Discoverer pipeline and considered 75% phosphorylation probability as an adequate threshold for phosphorylation site assignment. We performed the SEQUEST analysis against the SGD database, as well as a decoy database (reversed sequences), and calculated an empirical FDR < 1% at the level of peptide spectrum matches (PSMs). Separately, we calculated an FDR at peptide and protein level as well (FDR < 1%). The potential arginine-to-proline conversion was corrected by calculating a factor based on the SILAC ratio biases observed for peptide groups that are different in the number of prolines. SILAC Heavy-to-Light ratios were accordingly corrected and log2-transformed.

The mass spectrometry proteomics data have been deposited to the ProteomeXchange Consortium (http://proteomecentral.proteomexchange.org) via the PRIDE partner repository (Perez-Riverol et al, 2021) with the dataset identifier PXD028461.

## Data availability

The datasets produced in this study are available in the following database: Phospho-proteomic MS data: PRIDE PXD028461.

## Peer review information

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

## Acknowledgements

We thank Siri-Jasmin Schultz and Belen Pacheco for performing initial Pho8Δ60 experiments, Dan Klionsky for anti-Atg13 antibodies, and the Vienna Biocenter Core Facilities (VBCF) for providing the LC–MS instrument pool. The Kraft laboratory has received funding from the Deutsche Forschungsgemeinschaft (DFG, German Research Foundation); Project ID 409673687; SFB 1381 (Project ID 403222702); SFB 1177 (Project ID 259130777); under Germany's Excellence Strategy (CIBSS - EXC-2189- Project ID 390939984); form the Vienna Science and Technology Fund (WWTF, VRG10-001); from the European Research Council (ERC) under the European Union's Horizon 2020 research and innovation programme under grant agreement No 769065; from the European Union's Horizon 2020 research and innovation programme under grant agreement No 765912. NR, WR, and GA were supported by the FWF Austrian Science Fund Special Research Program F34. This work was further supported by ENW KLEIN-1 (OCENW.KLEIN.118), ZonMW TOP (91217002) SNSF Sinergia (CRSII5_189952) and Novo Nordisk Foundation (0066384) grants to FR. This work reflects only the authors' view and the European Union's Horizon 2020 research and innovation programme is not responsible for any use that may be made of the information it contains. The authors declare no competing financial interests. The Kraft laboratory thanks the SGBM graduate school for supporting their students. Work included in this study has also been performed in partial fulfillment of the requirements for the doctoral thesis of AB, ML, and AC at the University of Freiburg.

## Author contributions

**Anuradha Bhattacharya**: Validation; Investigation; Visualization; Methodology; Writing—review and editing. **Raffaela Torggler**: Validation; Investigation; Visualization. **Wolfgang Reiter**: Supervision; Validation; Investigation; Visualization; Writing—review and editing. **Natalie Romanov**: Validation; Investigation. **Mariya Licheva**: Validation; Investigation; Visualization. **Akif Ciftci**: Validation; Investigation; Visualization. **Muriel Mari**: Validation; Investigation; Visualization. **Lena Kolb**: Validation; Investigation. **Dominik Kaiser**: Validation; Investigation. **Fulvio Reggiori**: Supervision; Funding acquisition; Validation; Writing—review and editing. **Gustav Ammerer**: Supervision; Funding acquisition; Validation. **David M Hollenstein**: Conceptualization; Supervision; Validation; Visualization; Methodology; Writing—original draft; Project administration; Writing—review and editing. **Claudine Kraft**: Conceptualization; Supervision; Funding acquisition; Validation; Visualization; Methodology; Writing—original draft; Project administration; Writing—review and editing.

## Funding

## Disclosure and competing interests statement

The authors declare no competing interests.

