## [Peer Review File · EMBO Reports]

Decoding the function of Atg13 phosphorylation reveals a role of Atg11 in bulk autophagy initiation

Anuradha Bhattacharya, Raffaella Torggler, Wolfgang Reiter, Natalie Romanov, Mariya Licheva, Akif Ciftci, Muriel Mari, Lena Kolb, Dominik Kaiser, Fulvio Reggiori, Gustav Ammerer, David Hollenstein, and Claudine Kraft

DOI: [10.15252/embr.202357821](https://doi.org/10.15252/embr.202357821)

Corresponding authors: Claudine Kraft (claudine.kraft@biochemie.uni-freiburg.de) , David Hollenstein (david.hollenstein@univie.ac.at)

Review Timeline:

Submission Date:	14th Jul 23
Editorial Decision:	14th Aug 23
Revision Received:	17th Oct 23
Editorial Decision:	22nd Nov 23
Revision Received:	15th Dec 23
Accepted:	20th Dec 23

Transaction Report:

Dear Claudine,

Thank you for the submission of your research manuscript to our journal. We have now received the full set of referee reports that is copied below.

As you will see, the referees acknowledge that the findings are potentially interesting, but they also raise a number of concerns and have suggestions how to strengthen your study.

Given these constructive comments, we would like to invite you to revise your manuscript with the understanding that the referee concerns (as detailed above and in their reports) must be fully addressed and their suggestions taken on board. Please address all referee concerns in a complete point-by-point response. Acceptance of the manuscript will depend on a positive outcome of a second round of review. It is EMBO Reports policy to allow a single round of revision only and acceptance or rejection of the manuscript will therefore depend on the completeness of your responses included in the next, final version of the manuscript.

We realize that it is difficult to revise to a specific deadline. In the interest of protecting the conceptual advance provided by the work, we recommend a revision within 3 months (November 14th). Please discuss the revision progress ahead of this time with the editor if you require more time to complete the revisions.

I am also happy to discuss the revision further via e-mail or a video call, if you wish.

*****IMPORTANT NOTE:

We perform an initial quality control of all revised manuscripts before re-review. Your manuscript will FAIL this control and the handling will be DELAYED IF the following APPLIES:

- 1) A data availability section providing access to data deposited in public databases is missing. If you have not deposited any data, please add a sentence to the data availability section that explains that.
- 2) Your manuscript contains statistics and error bars based on $n=2$. Please use scatter blots in these cases. No statistics should be calculated if $n=2$.

When submitting your revised manuscript, please carefully review the instructions that follow below. Failure to include requested items will delay the evaluation of your revision.*****

- 1) a .docx formatted version of the manuscript text (including legends for main figures, EV figures and tables). Please make sure that the changes are highlighted to be clearly visible.
- 2) individual production quality figure files as .eps, .tif, .jpg (one file per figure). Please download our Figure Preparation Guidelines (figure preparation pdf) from our Author Guidelines pages <https://www.embopress.org/page/journal/14693178/authorguide> for more info on how to prepare your figures.
- 3) a .docx formatted letter INCLUDING the reviewers' reports and your detailed point-by-point responses to their comments. As part of the EMBO Press transparent editorial process, the point-by-point response is part of the Review Process File (RPF), which will be published alongside your paper.
- 4) a complete author checklist, which you can download from our author guidelines (<<https://www.embopress.org/page/journal/14693178/authorguide>>). Please insert information in the checklist that is also reflected in the manuscript. The completed author checklist will also be part of the RPF.
- 5) Please note that all corresponding authors are required to supply an ORCID ID for their name upon submission of a revised manuscript (<<https://orcid.org/>>). Please find instructions on how to link your ORCID ID to your account in our manuscript tracking system in our Author guidelines (<<https://www.embopress.org/page/journal/14693178/authorguide#authorshipguidelines>>)
- 6) We replaced Supplementary Information with Expanded View (EV) Figures and Tables that are collapsible/expandable online. A maximum of 5 EV Figures can be typeset. EV Figures should be cited as 'Figure EV1, Figure EV2' etc... in the text and their respective legends should be included in the main text after the legends of regular figures.

7) When referring to the mass spectrometry proteomics deposited to ProteomeXchange Consortium please add the header "Data Availability " and follow the model below (see also <<https://www.embopress.org/page/journal/14693178/authorguide#dataavailability>>).

Data availability

Additional information on source data and instruction on how to label the files are available <<https://www.embopress.org/page/journal/14693178/authorguide#sourcedata>>.

10) Figure legends and data quantification:

- The name of the statistical test used to generate error bars and P values,
 - the number (n) of independent experiments (please specify technical or biological replicates) underlying each data point,
 - the nature of the bars and error bars (s.d., s.e.m.)
- If the data are obtained from n {less than or equal to} 5, show the individual data points in addition to the SD or SEM.
- If the data are obtained from n {less than or equal to} 2, use scatter blots showing the individual data points.

11) All Materials and Methods need to be described in the main text. We would encourage you to use 'Structured Methods', our new Materials and Methods format. According to this format, the Materials and Methods section should include a Reagents and Tools Table (listing key reagents, experimental models, software and relevant equipment and including their sources and relevant identifiers) followed by a Methods and Protocols section in which we encourage the authors to describe their methods using a step-by-step protocol format with bullet points, to facilitate the adoption of the methodologies across labs. More information on how to adhere to this format as well as downloadable templates (.doc or .xls) for the Reagents and Tools Table can be found in our author guidelines: <

<https://www.embopress.org/page/journal/14693178/authorguide#manuscriptpreparation>>. An example of a Method paper with Structured Methods can be found here: <<https://www.embopress.org/doi/10.15252/msb.20178071>>.

12) Our journal encourages inclusion of *data citations in the reference list* to directly cite datasets that were re-used and obtained from public databases. Data citations in the article text are distinct from normal bibliographical citations and should directly link to the database records from which the data can be accessed. In the main text, data citations are formatted as

follows: "Data ref: Smith et al, 2001" or "Data ref: NCBI Sequence Read Archive PRJNA342805, 2017". In the Reference list, data citations must be labeled with "[DATASET]". A data reference must provide the database name, accession number/identifiers and a resolvable link to the landing page from which the data can be accessed at the end of the reference. Further instructions are available at <<https://www.embopress.org/page/journal/14693178/authorguide#referencesformat>>.

13) As part of the EMBO publication's Transparent Editorial Process, EMBO Reports publishes online a Review Process File to accompany accepted manuscripts. This File will be published in conjunction with your paper and will include the referee reports, your point-by-point response and all pertinent correspondence relating to the manuscript.

Kind regards,

Martina

Referee #1:

The study by Bhattacharya et al addresses the role of the post-translational modification (phosphorylation) of the pivotal autophagy protein Atg13 in the regulation of autophagy in the yeast *S. cerevisiae*. By combining mutational analysis with the functional autophagy and viability assays, the authors reveal the impact of Atg13 hyperphosphorylation (or the lack thereof) on the bulk autophagy process as well as on the viability of yeast under stress conditions (amino acid starvation). Unexpectedly, using hyperphosphorylation-mimicking Atg13 mutant constructs (which strongly inhibit autophagy) artificially fused to Atg1 (partial recovery of autophagy observed), they also discover a role for the selective autophagy-specific factor Atg11 in the bulk autophagy (the formation of the phagophore assembly site, PAS, at the vacuole).

This study provides a conceptual advancement in the field of autophagy, as it not only contributes additional (much more detailed) understanding of the Atg13-mediated bulk autophagy, but also shows the potential crosstalk between the selective autophagy (Atg11 binds key selective autophagy receptors identified in yeast to date (i.e., Atg19, Atg30, Atg32, Atg34, Atg36, Atg39, and Atg40) and the bulk autophagy, something that we lack good understanding of.

I recommend some improvements to the manuscript below in the hope that it will gain I quality and impact once published in EMBO Reports.

Specific comments

1. The manuscript would clearly benefit from a table summarizing the Atg13 mutants created and studied together with the observed functional outcomes (i.e., interaction with partner proteins, impact on bulk autophagy, PAS formation, yeast survival under stress conditions, etc.). The table would help organize the data vs. messages and make it a valuable reference material in the field of autophagy.

2. Please indicate also in the text in which mutant background (i.e., deltaAtg13) functional assessment of Atg13 phospho-mutants was performed. This information is found in the figure and the figure legend but should formally be available when reading the Results text.

3. Mutating up to 44x Ser residues in Atg13 to either Ala or Asp is a significant alteration of the primary sequence of the protein. At the very least, in the Discussion the authors should describe their experience with vs. expectation for the protein that underwent this intervention. What is known about the Atg13 secondary/tertiary structure (alpha helix-structured HORMA domain

vs. unstructured rest of Atg13)? How would introduction of negatively charged amino acids in the supposedly unstructured region impact it? Is there literature that helps support the work performed by the authors of this study?

4. Interestingly, under conditions of Atg13-44A expression, with overactive autophagy, stronger interaction between Atg11 and Atg1 was detected (Fig. 4B). One hypothesis would be that such artificially high autophagy in yeast could consume Atg11 required for the selective autophagy pathway (e.g., the Cvt pathway). It would be important to test this by looking at the abundance of the Atg11:Atg19 complex (Co-IP studies) as well as the levels of the productive Cvt substrate delivery. Such data would strengthen the core message of the study on the crosstalk between the bulk and selective autophagy mediated by Atg11. Is Atg11 part of the PAS, under the above conditions, as is the case in the Cvt pathway (Yorimitsu and Klionsky 2005 MollBiolCell)?

5. In Fig. 5 please provide a schematic/cartoon on the architecture of the stable Atg1-Atg13 protein fusion used in experiments.

6. Having control for the defective Cvt pathway (preApe 1 assay) in Atg11-deficient cells would be important when talking about the impact of Atg11 deficiency on the bulk autophagy under the conditions when Atg1-Atg13-44D/MD fusion was expressed (e.g., Fig. 6C).

Referee #2:

The manuscript of Bhattacharya et al. provides a comprehensive analysis of Atg13 phosphorylation under growth and starvation conditions. The protein can be phosphorylated at 48 sites. The majority of these sites is phosphorylated preferentially under growth conditions, some are phosphorylated under both conditions, and two sites (S344 and S496) are increasingly phosphorylated during starvation. Excluding S428 and S429 from the mutational analysis due to their hydrogen bonding with Atg17 (neither phosphodeficient, nor phosphomimetic mutant of these two sites is functional), the authors mutated all other sites to A or D and observed either hyperactive or inactive autophagy due to increased or decreased PAS formation, respectively. Interestingly, this was true under both growth (TORC1 active) and starvation (TORC1 inactive) conditions and required Atg11 for efficient PAS formation. Importantly, while phosphorylation events in Atg13 middle region regulated recruitment of both Atg1 and Atg17, the phosphorylation events at Atg13 C-terminus affected binding of only Atg1 allowing to decipher an important role of Atg1 recruitment in PAS formation. Overall, this is a high-quality mechanistic study of early events that lead to PAS formation with an important physiological insight that hyperactive PAS formation and autophagy decrease survival during starvation. It will be of great interest to the broad autophagy community and can be accepted for publication after addressing the following comments:

1) Introduction, paragraph 1, line 7: "macromolecules" are not exported back into the cytosol from the lytic compartment - their building blocks are.

2) Results, paragraph 3: the authors disclose that their Atg13-44D mutant is, in fact, the Atg13-44D-2A (has S344A and S496A). But it is not clear if Atg13-46D is also Atg13-46D-2A or not.

3) In the next paragraph, it is stated that the Atg13-44A mutant is, actually, the Atg13-44A-2D (has S344D and S496D). But it is not clear if Atg13-46A is also Atg13-46A-2D or not.

4) Same paragraph 4 and Figure 2B: the co-IP experiment in Figure 2B did not address the Atg13-Atg17 binding, as interpreted. It addressed the Atg1-Atg13 and Atg1-Atg17 binding. The main reason why Atg1 with Atg13-46A pulls down less Atg17 than Atg1 with Atg13-44A is that Atg1 pulls down less Atg13-46A than Atg13-44A in the first place. Are S428 and S429 of Atg13 known to be involved in the Atg1-Atg13 interaction? The data suggests that they are involved under starvation conditions (Figure 2B), but not under growth conditions (Figure 4B). How many times both experiments were repeated? Are these results on differential co-IP of Atg13-44A and Atg13-46A with Atg1 in starved (but not fed) cells reproducible?

5) Next paragraph 5 and Figure 2C: I disagree with the conclusion that "these eight sites are not sufficient to regulate Atg13 activity". What if they are sufficient if mutated same as in 44D-2A, meaning 7D-1A (has S496A), instead of 8D (has S496D). Did you try it? It would be a more fair comparison.

6) Figure 3D legend: "asterisks, autophagic bodies; #, lipid droplets; *, autophagosomes" - Did you use asterisks to label both autophagic bodies and autophagosomes?

7) Section titled "Atg13's C-terminal...": "(Figure 2B, 5C and 5D)" - reference to 5C and 5D is incorrect.

8) Same section: it is not clear if Atg13-MA and Atg13-MD constructs followed the same rule as Atg13-44A and Atg13-44D constructs and were the Atg13-MA-2D and Atg13-MD-2A instead. If not, the comparison of these constructs with Atg13-44A and Atg13-44D (which are, in reality, Atg13-44A-2D and Atg13-44D-2A) is incorrect. This is the same concern, as in points 2, 3 and 5 above. Overall, I suggest being more transparent about these 2A vs 2D differences between various constructs by specifying

them in all the relevant names (see also points 2 and 3 above) in both text and figures.

9) Figure 7 and Tables S1-S3 are not mentioned anywhere in the manuscript.

Referee #3:

Bhattacharya et al identified 48 potentially phosphorylated sites on Atg13 in cells. The authors focused on 44 sites and show that a phospho-inhibitory mutant activates autophagy while a phospho-mimetic mutant inhibits autophagy. Members of the Atg1 complex are known to be heavily regulated by phosphorylation and this study provides a general understanding of the impact of Atg13 phosphorylation during autophagy. This study is well designed mainly only lacking a better description of some aspects of the study. I only have minor comments, mostly involving textual edits.

Whilst the authors start the study with analysing mutants of Atg13 that include 44 or 46 phospho-site mutants, they subsequently shorten these mutants and separate them depending on their location within the protein. This is quite an important initial step to breakdown the relevance of these post-translational modifications. However, it is still possible that only a handful or even one or two of the 44 mutated sites mediate the autophagy regulatory activity of Atg13, potentially depending on their location within the secondary structure of the protein. Certainly, narrowing down such possible mutations is laborious. The authors should mainly just include a discussion of such possibilities.

One important aspect missing in this study is a clarification/specification of the mutated sites included within mutant 44, N-terminal, C-terminal, or Middle regions. A table specifying these sites would be informative. The authors identified 48 phosphorylation sites on Atg13, but they generated mutants that include 44 or 46 of these sites. What are the 2 sites that were not studied? It would be good to clarify this in the results description. If possible, the authors can also highlight sites phosphorylated by known kinases.

Further details of the MS analyses would be important to include: 1) any differences in the incidence of the phosphorylation event detect per Atg13 peptide; and 2) any peptides that were not identified by MS.

What is the rationale for mutating all sites irrespective of their phosphorylation status during autophagy induction?

Whilst fusing mutant CD to Atg1 can restore its defect in autophagy, doing the same experiment with mutant MD only does so partially. Is this due to a defect of MD to bind Atg17 or possibly other factors? A discussion of this would be helpful.

Figure 2B: Is the ability of the phospho-mimetic mutant 44D to bind Atg1 unaffected in this figure? This is in contrast to the data in Figures 4B and 5D. A clarification of this finding in the result section related to figure 2B would be important.

Can statistical analyses be added to some of the graphs in this study? This is especially important in figures like 2A when the authors mention some autophagic activity under basal conditions of mutant 44A, and figure 5B where changes are not very clear.

Figure 3D legend: indication of autophagosomes/autophagic body as asterisks/* is redundant.

In some parts, the manuscript is sparse in references to support statement related to previously published findings. It would be good for the authors to revise their references and support their statements better. For example, pg 8, paragraph starting with "Several autophagy factors".

It would be informative if in all IP blots, the bait being pulled down is specified in the figures.

Response to the points raised by the referees

We thank all three referees for their insightful comments, which helped us to substantially improve our manuscript. As explained in detail below, we have addressed all points.

Referee #1:

The study by Bhattacharya et al addresses the role of the post-translational modification (phosphorylation) of the pivotal autophagy protein Atg13 in the regulation of autophagy in the yeast *S. cerevisiae*. By combining mutational analysis with the functional autophagy and viability assays, the authors reveal the impact of Atg13 hyperphosphorylation (or the lack thereof) on the bulk autophagy process as well as on the viability of yeast under stress conditions (amino acid starvation). Unexpectedly, using hyperphosphorylation-mimicking Atg13 mutant constructs (which strongly inhibit autophagy) artificially fused to Atg1 (partial recovery of autophagy observed), they also discover a role for the selective autophagy-specific factor Atg11 in the bulk autophagy (the formation of the phagophore assembly site, PAS, at the vacuole).

This study provides a conceptual advancement in the field of autophagy, as it not only contributes additional (much more detailed) understanding of the Atg13-mediated bulk autophagy, but also shows the potential crosstalk between the selective autophagy (Atg11 binds key selective autophagy receptors identified in yeast to date (i.e., Atg19, Atg30, Atg32, Atg34, Atg36, Atg39, and Atg40) and the bulk autophagy, something that we lack good understanding of.

I recommend some improvements to the manuscript below in the hope that it will gain I quality and impact once published in EMBO Reports.

Specific comments

1. The manuscript would clearly benefit from a table summarizing the Atg13 mutants created and studied together with the observed functional outcomes (i.e., interaction with partner proteins, impact on bulk autophagy, PAS formation, yeast survival under stress conditions, etc.). The table would help organize the data vs. messages and make it a valuable reference material in the field of autophagy.

As suggested, we have now included a table on the mutants used in this study (new Figure 5A). We have also better clarified throughout the text what exact mutations these different mutants include or exclude.

2. Please indicate also in the text in which mutant background (i.e., deltaAtg13) functional assessment of Atg13 phospho-mutants was performed. This information is found in the figure and the figure legend but should formally be available when reading the Results text.

Multiple of the experiments actually have been done with plasmids first and then with integrations later, and the results were the same. We included the precise setup in the figure legend and also in the way we labelled the figures, but we omitted this information from the main results text on purpose, as it becomes confusing to read when multiple deletion backgrounds are used. We strongly prefer to leave it this way for the sake of readability. We have however inserted the following statement to clarify:

“Throughout this study, mutated versions of Atg13 were either stably integrated at the native ATG13 locus in the genome, or expressed from a centromeric plasmid containing the native ATG13 promoter in an atg13Δ deletion strain.”

3. Mutating up to 44x Ser residues in Atg13 to either Ala or Asp is a significant alteration of the primary sequence of the protein. At the very least, in the Discussion the authors should describe their experience with vs. expectation for the protein that underwent this intervention. What is known about the Atg13 secondary/tertiary structure (alpha helix-structured HORMA domain vs. unstructured rest of Atg13)? How would introduction of negatively charged amino acids in the supposedly unstructured region impact it? Is there literature that helps support the work performed by the authors of this study?

When we first created these mutants, we assumed that due to the many mutations, these proteins would misfold, and therefore the protein would be unstable. However, as shown in this work, both the D and A mutants are well-expressed and stable, which actually was a surprise to us. As they also mimic reciprocal states, we strongly believe that these proteins do resemble their native phosphorylated and non-phosphorylated states. The negative charge introduced by the aspartates should be similar to the negative charge introduced by phosphorylation, and therefore expected to influence the unstructured region in a similar manner. We explain this in the text, and also added a statement that these mutants are stable:

In the results part:

“Despite the many mutations, these mutants were stably expressed (Figure 2A).”

In the discussion:

“Despite the large number of mutated residues, both mutants were stably expressed.”

We furthermore moved the blots showing the stability of the mutants to the main figures.

4. Interestingly, under conditions of Atg13-44A expression, with overactive autophagy, stronger interaction between Atg11 and Atg1 was detected (Fig. 4B). One hypothesis would be that such artificially high autophagy in yeast could consume Atg11 required for the selective autophagy pathway (e.g., the Cvt pathway). It would be important to test this by looking at the abundance of the Atg11:Atg19 complex (Co-IP studies) as well as the levels of the productive Cvt substrate delivery. Such data would strengthen the core message of the study on the crosstalk between the bulk and selective autophagy mediated by Atg11. Is Atg11 part of the PAS, under the above conditions, as is the case in the Cvt pathway (Yorimitsu and Klionsky 2005 *MolBiolCell*)?

This is an interesting point raised by the reviewer. To test if bulk autophagy induced by Atg13-44A under nutrient-rich conditions depletes Atg11 for the Cvt pathway, we tested if the Cvt pathway was affected by the Atg13-44A mutant. In fact, Ape1 processing was at least as good as in Atg13-wt containing cells, rather even slightly enhanced, suggesting that Atg11 is not depleted and also not limiting in Atg13-44A expressing cells, and can function in both the Cvt pathway and bulk autophagy in parallel. We included this data in the revised Figure 4B.

5. In Fig. 5 please provide a schematic/cartoon on the architecture of the stable Atg1-Atg13 protein fusion used in experiments.

As suggested by the reviewer, we added a cartoon of the Atg1-Atg13 protein fusion, along with more detailed cartoons of the other mutants used.

6. Having control for the defective Cvt pathway (preApe 1 assay) in Atg11-deficient cells would be important when talking about the impact of Atg11 deficiency on the bulk autophagy under the conditions when Atg1-Atg13-44D/MD fusion was expressed (e.g., Fig. 6C).

As suggested by the reviewer, we checked CVT pathway function in the Atg1-1344D/MD fusion mutants. As expected, in atg11 Δ or atg19 Δ cells, the CVT pathway was nonfunctional for all fusion mutants. We included this data in the revised Figure 6C.

Referee #2:

The manuscript of Bhattacharya et al. provides a comprehensive analysis of Atg13 phosphorylation under growth and starvation conditions. The protein can be phosphorylated at 48 sites. The majority of these sites is phosphorylated preferentially under growth conditions, some are phosphorylated under both conditions, and two sites (S344 and S496) are increasingly phosphorylated during starvation. Excluding S428 and S429 from the mutational analysis due to their hydrogen bonding with Atg17 (neither phosphodeficient, nor phosphomimetic mutant of these two sites is functional), the authors mutated all other sites to A or D and observed either hyperactive or inactive autophagy due to increased or decreased PAS formation, respectively. Interestingly, this was true under both growth (TORC1 active) and starvation (TORC1 inactive) conditions and required Atg11 for efficient PAS formation. Importantly, while phosphorylation events in Atg13 middle region regulated recruitment of both Atg1 and Atg17, the phosphorylation events at Atg13 C-terminus affected binding of only Atg1 allowing to decipher an important role of Atg1 recruitment in PAS formation. Overall, this is a high-quality mechanistic study of early events that lead to PAS formation with an important physiological insight that hyperactive PAS formation and autophagy decrease survival during starvation. It will be of great interest to the broad autophagy community and can be accepted for publication after addressing the following comments:

1) Introduction, paragraph 1, line 7: "macromolecules" are not exported back into the cytosol from the lytic compartment - their building blocks are.

We thank the reviewer for noting this mistake and have changed it accordingly.

2) Results, paragraph 3: the authors disclose that their Atg13-44D mutant is, in fact, the Atg13-44D-2A (has S344A and S496A). But it is not clear if Atg13-46D is also Atg13-46D-2A or not.

The Atg13-46D is "Atg13-46D-2A". We have clarified this in the revised text and added a cartoon for all mutants used in the revised Figure 5A.

3) In the next paragraph, it is stated that the Atg13-44A mutant is, actually, the Atg13-44A-2D (has S344D and S496D). But it is not clear if Atg13-46A is also Atg13-46A-2D or not.

The Atg13-46A is "Atg13-46A-2D". We have clarified this in the revised text and added a supplementary table for all mutants used.

4) Same paragraph 4 and Figure 2B: the co-IP experiment in Figure 2B did not address the Atg13-Atg17 binding, as interpreted. It addressed the Atg1-Atg13 and Atg1-Atg17 binding. The main reason why Atg1 with Atg13-46A pulls down less Atg17 than Atg1 with Atg13-44A is that Atg1 pulls down less Atg13-46A than Atg13-44A in the first place. Are S428 and S429 of Atg13 known to be involved in the Atg1-Atg13 interaction? The data suggests that they are involved under starvation conditions (Figure 2B), but not under growth conditions (Figure 4B). How many times both experiments were repeated? Are these results on differential co-IP of Atg13-44A and Atg13-46A with Atg1 in starved (but not fed) cells reproducible?

Fujioka et al. identified S428 and S429 as amino acids required for hydrogen bonding with Atg17, and a S429A mutant showed a strong reduction in Atg17 binding (Fujioka Figure 5b). They also analyzed the S429A mutant in Atg1 binding but did not observe a reduction compared to Atg13wt (Fujioka Suppl Fig 5c). The S428A/S429A double mutant has not been analyzed. We only analyzed these sites in the context of the additional phosphorylation sites on Atg13. In our hands, indeed, the 46A mutant, which includes the 428/9AA mutation, reproducibly showed less co-precipitation with Atg1, whereas the 44A mutant, which doesn't include the 428/9 mutations, shows rather more co-precipitation with Atg1 than Atg13wt (our Figure 2B). Under nutrient-rich conditions, the 44A and 46A mutants assemble with Atg1 in a similar manner (our Figure 4B). These co-IPs in Figure 2B and 4B have been performed n=4 and the results are reproducible. We have included this information in the figure legend and more examples are shown below. However, as Atg17 binding is also affected by SS428/9AA mutations (Fujioka et al.), the observed decrease in Atg1-Atg13 co-precipitation could also result from a less stable PAS in general, which we have stated as such in the revised results text.

5) Next paragraph 5 and Figure 2C: I disagree with the conclusion that "these eight sites are not sufficient to regulate Atg13 activity". What if they are sufficient if mutated same as in 44D-2A, meaning 7D-1A (has S496A), instead of 8D (has S496D). Did you try it? It would be a more fair comparison.

We agree with the reviewer that the 7D1A (7SD1A) mutant is a fairer comparison and also constructed this mutant. We compared this mutant to the 8SD mutant, but both mutants showed similar Pho8Δ60 activity and were unable to mimic the autophagy-inhibited state. We therefore conclude that these eight sites are indeed not sufficient to regulate Atg13 activity. These findings are shown in the new Figure 2F.

6) Figure 3D legend: "asterisks, autophagic bodies; #, lipid droplets; *, autophagosomes" - Did you use asterisks to label both autophagic bodies and autophagosomes?

We thank the reviewer for noting this mistake, in fact, there are no autophagosomes, as they fuse very rapidly with the vacuole. All asterisks mark autophagic bodies inside the vacuole, we corrected the figure legend accordingly.

7) Section titled "Atg13's C-terminal...": "(Figure 2B, 5C and 5D)" - reference to 5C and 5D is incorrect.

We thank the reviewer for noting this. We corrected the reference to the figures.

8) Same section: it is not clear if Atg13-MA and Atg13-MD constructs followed the same rule as Atg13-44A and Atg13-44D constructs and were the Atg13-MA-2D and Atg13-MD-2A instead. If not, the comparison of these constructs with Atg13-44A and Atg13-44D (which are, in reality, Atg13-44A-2D and Atg13-44D-2A) is incorrect. This is the same concern, as in points 2, 3 and 5 above. Overall, I suggest being more transparent about these 2A vs 2D differences between various constructs by specifying them in all the relevant names (see also points 2 and 3 above) in both text and figures.

The MA and MD mutants are the "MA-2D" and "MD-2A", resembling the individual domains of the 44A mutant. We included a cartoon of these mutants clarifying this, as well as clarified in the text.

9) Figure 7 and Tables S1-S3 are not mentioned anywhere in the manuscript.

We have included the reference to this figure and the tables in the revised text.

Referee #3:

Bhattacharya et al identified 48 potentially phosphorylated sites on Atg13 in cells. The authors focused on 44 sites and show that a phospho-inhibitory mutant activates autophagy while a phospho-mimetic mutant inhibits autophagy. Members of the Atg1 complex are known to be heavily regulated by phosphorylation and this study provides a general understanding of the impact of Atg13 phosphorylation during autophagy. This study is well designed mainly only lacking a better description of some aspects of the study. I only have minor comments, mostly involving textual edits.

1. Whilst the authors start the study with analysing mutants of Atg13 that include 44 or 46 phospho-site mutants, they subsequently shorten these mutants and separate them depending on their location within the protein. This is quite an important initial step to breakdown the relevance of these post-translational modifications. However, it is still possible that only a handful or even one or two of the 44 mutated sites mediate the autophagy regulatory activity of Atg13, potentially depending on their location within the secondary structure of the protein. Certainly, narrowing down such possible mutations is laborious. The authors should mainly just include a discussion of such possibilities.

As suggested by the reviewer, we now discuss this aspect in the revised discussion:

"Whether all of the mutated residues in the middle and c-terminal region of Atg13 contribute to the regulation of bulk autophagy or if this function is governed only by a subset of the sites needs further investigation.

2. One important aspect missing in this study is a clarification/specification of the mutated sites included within mutant 44, N-terminal, C-terminal, or Middle regions. A table specifying these sites would be informative. The authors identified 48 phosphorylation sites on Atg13, but they generated mutants that include 44 or 46 of these sites. What are the 2 sites that were not studied? It would be good to clarify this in the results description. If possible, the authors can also highlight sites phosphorylated by known kinases.

As suggested by the reviewer, we better describe in the revised manuscript which mutant contains which sites. We also included a cartoon depicting these different mutants in the revised Figure 5A. As for most of the phosphorylation sites, it is not absolutely clear which kinase directly phosphorylates a specific site in vivo, we prefer to not state any connections to specific kinases, as this will need further investigations.

3. Further details of the MS analyses would be important to include: 1) any differences in the incidence of the phosphorylation event detect per Atg13 peptide; and 2) any peptides that were not identified by MS.

As stated in the data availability section, we have uploaded the MS results to the PRIDE repository. This includes an Excel file summarizing the MS results related to Atg13, containing information on the phosphorylation site and peptide level, including phosphorylated and unphosphorylated peptides. We thank the reviewer for pointing out that this information should be more easily accessible to the reader. We have therefore included the table as a supplement to our manuscript. Moreover, we have revised the info tab of the Excel file to make the content more accessible.

4. What is the rationale for mutating all sites irrespective of their phosphorylation status during autophagy induction?

We have previously shown that Atg1 shows different activities in a cell depending on its localization, i.e. is active on cargo but not in the cytosol (Torggler et al., Mol Cell 2016). It is therefore possible that also Atg13 shows different phosphorylation states at different locations in the cell. Moreover, as we analyze a mixture of cells that are in different stages of autophagy, dynamic and transient changes of phosphorylation might average out and therefore appear unaffected. In this study, we took a comprehensive approach by mimicking the most complete reciprocal states of autophagy. We therefore decided to include a rather extensive set of sites for our mutational analysis, rather than a conservative one. We have undertaken the first step of unraveling the effect of distinct sites (or rather groups of sites) by analyzing the effect of mutations on specific domains. After having characterized this situation, it would be interesting to determine the contribution of each individual site and correlate this with the observed amount of regulation between rich and starved. Unfortunately, this was far beyond the scope of the current study.

5. Whilst fusing mutant CD to Atg1 can restore its defect in autophagy, doing the same experiment with mutant MD only does so partially. Is this due to a defect of MD to bind Atg17 or possibly other factors? A discussion of this would be helpful.

As Atg1 only binds Atg17 via Atg13 (Figure 2C), and Atg13-MD shows reduced Atg17 binding but Atg13-CD can interact with Atg17 well (Figure 5C), the most likely reason for the autophagy defect of the Atg1-13MD mutant is its inability to interact with Atg17. We have stated in the text, that the 13CD mutant defect only stems from its inability in binding to Atg1, as the Atg1-13CD fusion protein is functional, as well as that the 13MD mutant rescue was surprising, as it doesn't bind to Atg17:

"Indeed, the stable fusion of Atg1 with Atg13wt fully restored autophagy in atg1Δ atg13Δ cells, similar to the rescue of atg1Δ cells by Atg1 expression (Figure 5E)."

"The amount of restored autophagy activity was surprising, given that Atg13^{44D} and Atg13^{MD} showed hardly any Atg17 binding (Figure 5C)."

Further interactors of Atg13 are Atg9 and Vac8. From our FM analysis it is however clear, that PAS formation is affected, strongly suggesting that early factors such as Atg17 are affected. If other interactors such as Atg9 and Vac8 are also affected by phosphorylation, has not been tested.

6. Figure 2B: Is the ability of the phospho-mimetic mutant 44D to bind Atg1 unaffected in this figure? This is in contrast to the data in Figures 4B and 5D. A clarification of this finding in the result section related to figure 2B would be important.

The 44D mutant binds less to Atg1 in all experiments performed (n=4). In Figure 2B it might be less well visible due to some more Atg1-protA precipitated in the sample containing 44D. For clarity, we placed below a blot just with lanes 1 and 6, to facilitate comparison, as well as show two further replicates.

7. Can statistical analyses be added to some of the graphs in this study? This is especially important in figures like 2A when the authors mention some autophagic activity under basal conditions of mutant 44A, and figure 5B where changes are not very clear.

As suggested by the reviewer, we have included statistics for the less clear differences for Figures 2B (previously Figure 2A) and 5B and stated the *p* values in the results text. In addition, all source data has been included, allowing all readers to inspect and further analyze the data.

8. Figure 3D legend: indication of autophagosomes/autophagic body as asterisks/* is redundant.

We thank the reviewer for noting this mistake, in fact, there are no autophagosomes, as they fuse very rapidly with the vacuole. All asterisks mark autophagic bodies inside the vacuole, we corrected the figure legend accordingly.

9. In some parts, the manuscript is sparse in references to support statement related to previously published findings. It would be good for the authors to revise their references and support their statements better. For example, pg 8, paragraph starting with "Several autophagy factors".

As suggested, we have added more references to the primary literature in the revised manuscript.

10. It would be informative if in all IP blots, the bait being pulled down is specified in the figures.

We have added this information to all IP blots as suggested.

Manuscript number: EMBOR-2023-57821V2

Title: Decoding the function of Atg13 phosphorylation reveals a role of Atg11 in bulk autophagy initiation

Author(s): Claudine Kraft, Anuradha Bhattacharya, Raffaella Torggler, Wolfgang Reiter, Natalie Romanov, Mariya Licheva, Akif Ciftci, Muriel Mari, Lena Kolb, Dominik Kaiser, Fulvio Reggiori, Gustav Ammerer, and David Hollenstein

Dear Claudine,

Thank you for the submission of your revised manuscript to EMBO reports. I already informed you about the positive reports from the referees who were asked to assess it (copied below) and I am now writing with a formal 'accept in principle' decision, which means that I will be happy to accept your manuscript for publication once a few minor issues/corrections have been addressed, as follows.

- Please reduce the number of keywords to 5.
- Please update the 'Conflict of interest' paragraph to our new 'Disclosure and competing interests statement'. For more information see <https://www.embopress.org/page/journal/14693178/authorguide#conflictsofinterest>
- Please remove the Author Contributions from the manuscript file and make sure that the author contributions in our online submission system are correct and up-to-date. The information you specified in the system will be automatically retrieved and typeset into the article. You can enter additional information in the free text box provided, if you wish.
- "Methods" should be "Materials and Methods"
- Tables should be placed between main and EV figure legends
- Please describe all new findings in the abstract in present tense.
- A short side note: please remember to remove the reviewer access information from the Data availability section.
- Finally, EMBO Reports papers are accompanied online by A) a short (1-2 sentences) summary of the findings and their significance, B) 2-3 bullet points highlighting key results and C) a synopsis image that is 550x300-600 pixels large (width x height) in PNG for JPG format. You can either show a model or key data in the synopsis image. Please note that the size is rather small and that text needs to be readable at the final size. Please send us this information along with the revised manuscript.

If all remaining corrections have been attended to, you will then receive an official decision letter from the journal accepting your manuscript for publication in the next available issue of EMBO reports. This letter will also include details of the further steps you need to take for the prompt inclusion of your manuscript in our next available issue.

Thank you for your contribution to EMBO reports.

Kind regards,

Martina

Referee #1:

The authors have responded to all the points I raised during my initial review. I am satisfied with the quality of the manuscript and would accept it for publishing in EMBO Reports.

Referee #2:

The authors have dealt adequately with my concerns.

The authors addressed the minor editorial issues.

Prof. Claudine Kraft
University of Freiburg
Institute for Biochemistry and Molecular Biology, ZBMZ
Stefan-Meier Strasse 17
Freiburg
Germany

Dear Claudine,

I am very pleased to accept your manuscript for publication in the next available issue of EMBO reports. Congratulations to a successful publication and thank you for your contribution to our journal.

Best wishes,

Martina
